# CMRLCCOA: Multi-Strategy Enhanced Coati Optimization Algorithm for Engineering Designs and Hypersonic Vehicle Path Planning

**DOI:** 10.3390/biomimetics9070399

**Published:** 2024-07-01

**Authors:** Gang Hu, Haonan Zhang, Ni Xie, Abdelazim G. Hussien

**Affiliations:** 1Department of Applied Mathematics, Xi’an University of Technology, Xi’an 710054, China; zhanghaonan2023@163.com (H.Z.); xieni2024@sina.com (N.X.); 2School of Computer Science and Engineering, Xi’an University of Technology, Xi’an 710048, China; 3Department of Computer and Information Science, Linköping University, 58183 Linköping, Sweden; abdelazim.hussien@liu.se; 4Faculty of Science, Fayoum University, Faiyum 63514, Egypt; 5Applied Science Research Center, Applied Science Private University, Amman 11931, Jordan; 6MEU Research Unit, Middle East University, Amman 11831, Jordan

**Keywords:** coati optimization algorithm, chaos mapping strategy, Lévy flight strategy, lens imaging reverse learning strategy, crossover strategy

## Abstract

The recently introduced coati optimization algorithm suffers from drawbacks such as slow search velocity and weak optimization precision. An enhanced coati optimization algorithm called CMRLCCOA is proposed. Firstly, the Sine chaotic mapping function is used to initialize the CMRLCCOA as a way to obtain better-quality coati populations and increase the diversity of the population. Secondly, the generated candidate solutions are updated again using the convex lens imaging reverse learning strategy to expand the search range. Thirdly, the Lévy flight strategy increases the search step size, expands the search range, and avoids the phenomenon of convergence too early. Finally, utilizing the crossover strategy can effectively reduce the search blind spots, making the search particles constantly close to the global optimum solution. The four strategies work together to enhance the efficiency of COA and to boost the precision and steadiness. The performance of CMRLCCOA is evaluated on CEC2017 and CEC2019. The superiority of CMRLCCOA is comprehensively demonstrated by comparing the output of CMRLCCOA with the previously submitted algorithms. Besides the results of iterative convergence curves, boxplots and a nonparametric statistical analysis illustrate that the CMRLCCOA is competitive, significantly improves the convergence accuracy, and well avoids local optimal solutions. Finally, the performance and usefulness of CMRLCCOA are proven through three engineering application problems. A mathematical model of the hypersonic vehicle cruise trajectory optimization problem is developed. The result of CMRLCCOA is less than other comparative algorithms and the shortest path length for this problem is obtained.

## 1. Introduction

An optimization problem is to achieve the optimal value of the design objective under certain constraints. Optimization problems exist widely in intelligent production, engineering manufacturing, agricultural development, and many other fields [1]. But as the rapidly evolving digital age, data are showing explosive growth, and there are more and more multidimensional and multimodal problems, making many real-world problems more complex and diverse [2]. For traditional mathematical optimization means, such as the gradient descent method [3], conjugate gradient method [4], and quasi-Newton method [5], they often have some limitations when handling both discrete and other questions [6]. They have a tendency to trap into local optimal solutions, slow convergence speed, or low computational accuracy. Therefore, it is very difficult to use traditional algorithms for calculation and extracting meaningful information. Generally speaking, solving NP hard problems involves finding a point in a multidimensional hyperspace, which is the optimal solution. However, the identification process is very complex, time-consuming, and computationally expensive. Therefore, it is important to find a biomimetic computing method that is fast and effective [1].

Nowadays, metaheuristic algorithms have been proven to be competitive alternative algorithms, often used to solve highly complicated nonlinear optimization issues, such as multi-objective optimization problems [7], multimodal optimization problems [8], and complex constraint optimization problems [9].

Metaheuristic algorithms can avoid local optima and have faster convergence speed, better robustness, and higher stability than traditional algorithms [10]. This field has been developed so far that quite a number of very classical algorithms have been proposed. This mainly includes a genetic algorithm (GA) that emulates the evolutionary processes of living species [11], differential evolution (DE) that optimizes a search through cooperation and competition among individuals within a group [12], artificial immune systems (AIS) for modeling the body immune response mechanism [13], an ant colony algorithm (ACO) using ants’ way finding behavior as a model [14], particle swarm optimization (PSO) modeling the actions of birds in search of food [15], a simulated annealing algorithm (SA) modeled on the annealing procedure with solid materials [16], and a taboo search algorithm (TSA) for modeling the human intellectual memory procedure [17]. They always emerge from the imitation or exposition of specific natural occurrences and sequences, or the cognitive actions of living collectives and have the characteristics of simplicity, universality, and ease of parallel processing.

Based on sources of inspiration, metaheuristic algorithms mainly include evolution algorithms, human behavior-based algorithms, physics and chemistry-based algorithms, and swarm intelligence-based algorithms [18,19].

The evolutionary algorithm is based on concepts such as biology and genetics and is built by modeling the laws of nature’s superiority and inferiority. They achieve population progress according to the laws of natural selection, and thus finish the best solution. Conventional evolutionary algorithms are primarily represented by GA and DE. Both algorithms are modeled from the principle of reproduction in nature and then use strategies such as crossover, selection, and mutation to update the population.

Human behavior-based algorithms are inspired by human performance, such as self-learning actions and social activities [20]. The most commonly used algorithm is the imperialist competition algorithm [21], social-based algorithms [22], league championship algorithm [23], and poor and rich optimization algorithm [24].

The algorithms based on physics and chemistry mainly come from the physical laws and chemical phenomena in the universe. Among them, SA mentioned above is a classical algorithm. Furthermore, there are many algorithms developed from physical laws, such as gravity search algorithms [25] based on the law of universal gravitation; chaos optimization algorithms [26] based on the traversal, randomness, and regularity of chaotic phenomena; optical optimization algorithms [27] based on the principle of optical reflection; and black hole algorithms based on strong attractive forces [28].

Swarm intelligence-based algorithms simulate the behavior of natural populations such as ants, birds, bees, whales, lions, wolves, etc. Each population is a population of organisms. Groups search for the best location among themselves through behaviors such as cooperation and hunting. The representative algorithms are PSO and ACO referred to above. In addition, there are many algorithms of this type, such as beluga whale optimization [29], grey wolf optimizer [30], marine predator algorithm [31], white shark optimizer [32], emperor penguin optimizer [33], and so on.

For metaheuristic algorithms, the ability to explore and develop directly determines the performance of the algorithms [34]. Their imbalance will directly cause a reduction in the precision of problem-solving. Weak exploration ability will affect the population to explore more places, which will lead to getting trapped at local optima. And the lack of exploitation ability will directly affect the population’s ability to find the optimal value. This is likewise a prevalent issue with current optimization methodologies. At the same time, this is exactly what is meant by improvements to the algorithm.

M. Dehghani et al. [35] presented the coati optimization algorithm (COA) in 2023. Coatis are very active, agile in movement, and have strong adaptability. They forage during the day and rest on trees at night. Iguanas are one of the favorite foods of long coatis, and coatis often cooperate to prey on them. In addition, long coatis also face the risk of being preyed upon. Thus, COA was inspired by the strategies used by long-haired coatis when they are attacking iguanas, as well as the strategies they use when facing and avoiding predators. Although COA shows a high level of competitiveness on some of the problems, it still has room for improvement. According to the literature [36], COA always exhibits a state of premature convergence and is highly susceptible to falling into a local optimum. Meanwhile, during the experiments, the performance of COA relative to some newly proposed superior metaheuristic algorithms is always at a disadvantage when facing large-scale problems. And the disadvantage of low diversity in COA populations cannot be ignored either. As a result, many researchers have enhanced COA to solve more sophisticated engineering problems. F.A. Hashim et al. [37] proposed an efficient adaptive mutation COA and applied it to feature selection and global optimization. P. Tamilarasu and G. Singaravel [38] use an improved COA to achieve efficient scheduling in cloud computing environments. K. Thirumoorthy and J.J.B. J [39] improved the COA and applied it to breast cancer classification.

Nevertheless, the No Free Lunch derivation [40] has indicated that no single algorithm is capable of addressing every optimization challenge flawlessly. Excellent performance on one problem may not lead to a viable solution on another unrelated problem. As a result, researchers need to constantly develop new algorithms or make targeted improvements to certain algorithms to cope with increasingly complex real-world problems. Therefore, the improvement in some existing algorithms is very necessary.

Consequently, in this paper, the chaotic mapping strategy, lens imaging reverse learning strategy, crossover strategy, and Lévy flight strategy are applied to improve the COA. Firstly, the chaotic mapping strategy [1] is introduced in the population initialization stage to use chaotic sequences for population initialization to obtain higher-quality populations. Secondly, the use of the lens imaging reverse learning strategy [41] not only improves population diversity but also enlarges the scope of the search. In the early stage, the Lévy flight strategy [42] is applied. It allows the population to get rid of partial optima and expand the search capability. In the end, the introduction of the crisscross optimization algorithm [43] helps to amend the phenomenon of early convergence of the algorithm. The amalgamation of these strategies augments the optimization capability of the COA. The innovations as well as the main contributions of this paper are as follows.

(1)The enhanced COA consists of four strategies, namely the chaotic mapping strategy, the lens imaging reverse learning strategy, the Lévy flight strategy, and the crossover strategy.(2)The effect of 10 common chaotic strategies to improve the COA is analyzed and the optimal strategy is finally selected.(3)The CMRLCCOA is compared with the primitive COA, six new algorithms proposed in the last two years, four classic and well-recognized algorithms, and three improved algorithms, which are tested with the functions included in the CEC2017 and CEC2019 function sets. In addition, *dim* = 50 and 100 were also selected in the CEC2017 test set.(4)CMRLCCOA is used to solve three engineering optimization problems, including a single-stage cylindrical gear reducer, a welded beam design problem, and a cantilever beam design problem.(5)This paper establishes a mathematical model of the cruise trajectory of a hypersonic vehicle and solves the path planning problem with the newly proposed CMRLCCOA. Furthermore, the results of nine algorithms are compared. Thus, the reliability of CMRLCCOA is verified.

The remainder of this paper is organized as follows: Section 2 briefly describes the mathematical model of COA. Section 3 describes the detailed structure of the CMRLCCOA algorithm. The performance of the CMRLCCOA is evaluated on the basis of numerical experimental results in Section 4. Section 5 solves three real-world problems using CMRLCCOA. In Section 6, The cruise ballistic trajectory problem for hypersonic vehicles is modeled and solved using CMRLCCOA. Finally, Section 7 summarizes this paper.

## 2. Introduction to Mathematical Modeling of Coati Optimization Algorithm

The coati optimization algorithm (COA) is a new metaheuristic algorithm proposed in 2023 [35]. It is inspired by two natural behaviors of the coatis, including strategies for cooperating in attacking iguanas and behavioral strategies for facing and escaping predators. In p-dimensional space, each coati acts as a separate searching individual. The hunting process and escape process of the coati from predators are both individual updates. The position of the coati will be dynamically adjusted according to the position of the iguana and the migrated locations, and finally the globally optimal location (the best candidate solution) will be selected. Next, we will briefly introduce the mathematical model of COA.

### 2.1. Initialization Process

Firstly, the COA initializes m random individuals, $x_{1},\ldots ,x_{i},\ldots x_{m}$, by Equation (1), where the maximum boundary for individual values is $X_{\max}=(x_{1}^{\max},\ldots ,x_{p}^{\max})$ and the minimum boundary is $X_{\min}=(x_{1}^{\min},\ldots ,x_{p}^{\min})$. Then, evaluate the initialized random individuals through the objective function.
(1)$$x_{i,j}=x_{j}^{\min}+rand\cdot (x_{j}^{\max}-x_{j}^{\min}),\;i=1,2,\ldots m,\;j=1,2,\ldots ,p.$$

### 2.2. Hunting and Attack Strategies (Exploration Phase)

Coatis attack iguanas in groups. In this strategy, the coatis are first divided into two groups. One group climbs up a tree to approach and scare the iguana, while the other group waits quietly on the ground. When the iguana drops, the raccoons quickly attack and catch it. Figure 1 shows coatis attacking an iguana.

There are two assumptions in this strategy. First, the iguana is assumed to be in the optimal position. Second, half of the species climbed trees and half waited on the ground. Equation (2) is used to simulate the process of coatis climbing trees.
(2)$$x_{i,j}^{new}=x_{i,j}+b\cdot (Igu_{j}-I\cdot x_{i,j}),\;i=1,2,\ldots ,\left\lfloor \frac{m}{2}\right\rfloor .$$

When the iguana is frightened to land, the position of the iguana is set randomly; however, the other half of the coatis waiting on the ground move according to the random placement of the iguana. This behavior is represented by Equations (3) and (4). A schematic of the iguana’s updated position is shown in Figure 2.
(3)$$Igu_{j}=x_{j}^{\min}+b\cdot (x_{j}^{\max}-x_{j}^{\min}),\;j=1,2,\ldots ,p,$$
(4)$$x_{i,j}^{new}=\left\{\begin{array}{l} x_{i,j}+b\cdot (Igu_{j}-I\cdot x_{i,j}),\;F_{Igu}<F_{i}\\ x_{i,j}+b\cdot (x_{i,j}-Igu_{j}),\;F_{Igu}>F_{i} \end{array}\right.,\;i=\left[\frac{N}{2}\right]+1,\left[\frac{N}{2}\right]+2,\ldots ,N,\;j=1,2,\ldots ,m,$$
where $x_{i,j}$ represents the *j*-th dimension of the *i*-th coati, and *b* is a stochastic number between [0, 1]. *Igu* is the randomly given location of the iguana. *I* is any stochastic value in 1 and 2.

### 2.3. The Stage of Escaping Predators (Exploitation Stage)

During the development phase, the strategy adopted by the coati in facing and escaping predators is used to update its position. When a predator captures a coati, the coati quickly runs away and enters a relatively safe position, approaching the optimal position, as shown in Figure 3. This strategy demonstrates the capability of the COA algorithm development.

The strategy during the development phase is simulated using Equations (5) and (6).
(5)$$low_{j}=\frac{x_{j}^{\min}}{t},\;up_{j}=\frac{x_{j}^{\max}}{t},t=1,2,...,T,$$
(6)$$x_{i,j}^{new}=x_{i,j}+(1-2b)\cdot (low_{j}+b\cdot (up_{j}-low_{j})),\;i=1,2,...,m;\;j=1,2,...,p$$
where *low_j_* and *up_j_* indicate the local lower and upper bound of the *j*-th dimension, a stochastic number with a value of b between [0, 1].

If the updated adaptation value for the coati is preferable to the original adaptation value, the value is accepted; otherwise, do not accept this value. Equation (7) represents the update process.
(7)$$X_{i}=\left\{\begin{array}{l} X_{i}^{new},\;F_{i}^{new}<F_{i}\\ X_{i},\;\mathrm{else} \end{array}\right..$$

## 3. Multi-Strategy Enhanced COA

This part uses four strategies to strengthen COA. These strategies are the chaotic mapping strategy, lens imaging reverse learning strategy, Lévy flight strategy, and crossover strategy. The newly proposed CMRLCCOA solves the shortcomings of COA, which is prone to local optimization and premature convergence. The improvement strategy of the algorithm is presented next and the results are briefly analyzed.

### 3.1. Chaos Mapping Strategy

The traditional COA adopts the method of randomly setting the initial population, which is difficult to spread throughout the population, resulting in a lack of diversity in the original coati population and restricting the flexibility. Chaos mapping was first proposed by Lorenz et al. in 1963 [44]. Chaos mapping has characteristics such as randomness, traversal, and regularity [45]. This strategy can guarantee the diversity of the original population. Therefore, many intelligent algorithms employ chaotic mapping strategies to strengthen the optimization of algorithms. Zeng et al. use chaotic mapping to generate a random and regular initialization particle swarm, improving global search capability [46]. Xin et al. applied the chaotic mapping method for reinforcing the sparrow optimization algorithm [47].

Chaos theory mainly studies the behavior of dynamic systems that are sensitive to initial states. The method of generating an initial population through chaotic mapping is to first use a one-dimensional chaotic map, specify a random initial value in it, and iteratively generate a series of continuous points. Chaotic mapping strategies can boost the competence of population diversity, success rate, and convergence. Table 1 describes ten common chaotic mapping functions. To have a clearer perception of these functions, Figure 4 visualizes some of the initialization functions. The image shows that these mappings allow random initial population positions to be evenly distributed in the search space. For the selection of different initialization methods, please see Section 4.3.

### 3.2. Reverse Learning Strategy for Lens Imaging

Many intelligent optimization algorithms have low population diversity in the later stages of the iteration and do not easily search for optimal solutions. It is difficult to jump out when a coati searching for an individual falls into a local optimum. Consequently, a specular reflection learning strategy is introduced in this paper. This strategy is an optimization mechanism [58], which extends the algorithm’s search area by computing the inverse solution at the current position. Therefore, it increases the likelihood of discovering the ideal solution. However, reverse learning strategies need to be combined with the principle of lens imaging to achieve better results [59].

Imaging by a convex lens is an optical law. A convection lens has an object and a solid image on each side of the lens. The diagram is depicted in Figure 5.

The lens imaging formula can be derived from Figure 5 as follows:(8)$$\frac{1}{u}+\frac{1}{v}=\frac{1}{f}$$
where *u* is the object distance, *v* is the distance imaged, and *f* is the focal length.

For the reverse learning strategy of convex lens imaging, an individual *P* is imaged in a convex lens as in one-dimensional space. This is shown in Figure 6. The principle of lens imaging is expressed as Equation (9).
(9)$$X^{\prime }=\frac{p+q}{2}+\frac{p+q}{2k}-\frac{X}{k}.$$

Equation (10) is the solution formula for reverse learning of lens imaging, which is extended to *D*-dimensional optimization problems. The reverse learning formula based on lens imaging is obtained as follows:(10)$${X^{\prime }}_{j}=\frac{p_{j}+q_{j}}{2}+\frac{p_{j}+q_{j}}{2k}-\frac{X_{j}}{k}.$$

Among them, *p_j_* is the minimum in the *j*-th dimension, and *q_j_* is the maximum in the *j*-th dimension. *X_j_’* and *X_j_* are the inverse solutions of the lens.

### 3.3. Lévy Flight Strategy

In the COA, the position update is highly influenced by the iguana. However, the position update range of iguanas is small, so the search space and solution space of this algorithm are limited. The Lévy flight strategy is a stochastic behavior strategy proposed by Paul Lévy in 1937 [60], used to simulate the step size and direction during random walking or search processes. In this paper, the Lévy flight strategy is incorporated into the search phase of the COA to enlarge the search scope. Figure 7 depicts the Lévy distribution along with their trajectories in two- and three-dimensional spaces. This random wandering behavior can be effective in increasing the diversity of populations, which in turn allows individuals to explore a wider range of space. Then, the Lévy flight process can be described as a random walk process, as shown in Equation (11).
(11)$$\mathrm{Levy}(\lambda )\sim u=t^{-1-\lambda },\;0<\lambda \leq 2.$$

The *λ* can be calculated using the Mantegna method, as shown in Equation (12).
(12)$$s=\frac{\mu }{{\left|v\right|}^{\frac{1}{\lambda }}},$$
where *λ* is set to be 1.5, and *μ* and *v* follow a normal distribution.
(13)$$\sigma_{v}=1,$$
(14)$$\sigma_{\mu }={\left[\frac{\Gamma (1+\lambda )\cdot \sin(\pi \lambda /2)}{\lambda \cdot \Gamma \left[(1+\lambda )/2\right]\cdot 2^{(\lambda -1)/2}}\right]}^{\frac{1}{\beta }},$$
where Γ is the gamma function.

Therefore, Equation (15) is utilized to change the position of the coati.
(15)$${\bar{x}}_{t+1}={\bar{x}}_{t}^{rand}+\alpha \cdot \mathrm{sign}(rand-1/2)⊕\mathrm{Levy}(\lambda ),$$
where sign(*rand* – ½) can take three values, namely −1, 0, or 1. *α* represents the control quantity of step length, which can be expressed using Equation (16).
(16)$$\alpha =\alpha_{0}\left\lfloor {\bar{x}}_{t}^{rand}-{\bar{x}}_{t}\right\rfloor ,$$
where *α_0_* is set to be 0.01.

Then, Equation (15) can be represented as
(17)$${\bar{x}}_{t+1}={\bar{x}}_{t}^{rand}+\alpha_{0}\cdot \mathrm{sign}(rand-0.5)\cdot \frac{\mu }{{\left|v\right|}^{\frac{1}{\lambda }}}\cdot ({\bar{x}}_{t}^{rand}-{\bar{x}}_{t}).$$

### 3.4. Cross Optimization Algorithm

Meng et al. proposed the crisscross optimization algorithm (CSO) [43]. The algorithm utilizes horizontal and vertical crossing to update information, which can effectively solve the local optimization problem.

#### 3.4.1. Horizontal Crossover

Before performing the crossover operation, two individuals are paired. Subsequently, the crossover is performed on the variables in the corresponding dimensions to generate new offspring. Assuming the *m*-th and *n*-th individuals are paired, the crossover operation is performed as follows:(18)$$Mhc_{m,j}=r_{1}\cdot X_{m,j}+(1-r_{1})\cdot X_{n,j}+c_{1}\cdot (X_{m,j}-X_{n,j}),$$
(19)$$Mhc_{n,j}=r_{2}\cdot X_{n,j}+(1-r_{2})\cdot X_{m,j}+c_{2}\cdot (X_{n,j}-X_{m,j}),$$
where *Mhc_m,j_* and *Mhc_n,j_* are descendants of *X_m,j_* and *X_n,j_*, respectively. And *X_m,j_* and *X_n,j_* are two random individuals in the population. *r*_1_ and *r*_2_ are randomly distributed evenly between 0 and 1. *c_1_* and *c_2_* are randomly distributed evenly between −1 and 1.

The first term in Equations (18) and (19) represents the particle’s current optimum, and the second term represents the mutual influence between two different particles, and these two terms are well combined through the weight factor *r*_1_. The third term can increase the search interval. The final solutions *Mhc_m,j_* and *Mhc_n,j_* must be compared with the fitness of the parent particles *X_m,j_* and *X_n,j_*, and the solution with better fitness should be retained for the next iteration.

#### 3.4.2. Vertical Crossover

Vertical crossover is executed across distinct dimensions of the variable. Due to the different ranges of values for different dimensions, they need to be normalized before crossing. Each vertical crossover only generates one offspring, and only updates one dimension of it.
(20)$$Mvc_{m,d_{1}}=r\cdot X_{m,d_{1}}+(1-r)\cdot X_{m,d_{2}},$$
where *r* is randomly distributed evenly between 0 and 1.

Vertical crossing can cause the dimension that has already fallen into a local optimum to escape from local optimality without damaging the information of the other dimension. Thus, in general, this strategy is effective in keeping population sizes from dropping into local minima, and the probability of a vertical crossover is lower than the probability of a horizontal crossover.

#### 3.4.3. Competitive Operator

There is a competitive relationship between the offspring population and the parent population. Only if the adaptation value of the offspring population is preferred to that of the parent population will it be retained and proceed to the next iteration. Otherwise, the parent population will continue to be retained. As a result of this simple competitive mechanism, individuals will move rapidly toward the search space with good fitness, close to the optimal solution. For example, in terms of horizontal crossing, the competition operator is defined as
(21)$$X_{m}^{offspring}=\left\{\begin{array}{l} X_{m},\;\;f(X_{m})<f(Mhc_{m})\\ Mhc_{m},\;\mathrm{else} \end{array}\right..$$

### 3.5. The Framework of CMRLCCOA

Inspired by the above strategies (chaotic mapping, lens imaging reverse learning, Lévy flight, and crossover strategy), we propose a new hybrid metaheuristic algorithm, CMRLCCOA. These strategies greatly strengthen the stability and optimization capability of the algorithm. The specific steps for solving the D-dimensional minimum problem using CMRLCCOA are as follows:

Step 1: Initialize some parameters of CMRLCCOA—the number of search agents *N*, dimension of the solution *D*, boundaries of variables *ub* and *lb*, and number of iterations *M_iter_*.

Step 2: Initializing *N* populations of coatis using chaotic mapping.

Step 3: The fitness values for each candidate solution are computed. Afterwards, record the best fitness value *f_best_* and the optimal position *X_best_*.

Step 4: Using a convex lens imaging reverse learning strategy to update *N* initial solutions by Equation (11), then calculating fitness values while retaining good fitness values and optimal solutions.

Step 5: While *C_iter_* < *M_iter_*, update the location of the iguana.

Step 6: For the first half of the individual coatis, using Equation (2) to change location of the *i*-th coati, and using Equation (7) again to update the position of the *i*-th coati.

Step 7: For the latter half of the individual coati, first set the iguana’s random location using Equation (3), then use Equation (4) to compute the new position of the *i*-th coati, and finally use Equation (7) to update the position of the *i*-th coati.

Step 8: Utilizing the Lévy flight strategy, the coati’s position is updated by Equation (18) and candidate solutions are calculated, while retaining the optimal solution and corresponding position.

Step 9: In the second stage of exploitation, first calculate the local boundaries of variables by Equation (5). The location of the *i*-th coati is changed using Equation (6). Equation (7) is used to update the optimal solution.

Step 10: Using the cross optimization strategy, horizontal and vertical crosses are performed on individuals of the coati by Equations (19)–(21) and offspring populations are obtained, and then the better preserved ones are selected from the parent and offspring populations.

Step 11: Set *C_iter_* = *C_iter_* + 1; if *C_iter_* < *M_iter_*, return to Step 5. Otherwise, the optimal location and fitness values obtained from solving the problem will be output.

To show the structure of the CMRLCCOA more clearly, the flowchart of CMRLCCOA is illustrated in Figure 8. Additionally, the pseudo-code of CMRLCCOA is shown in Algorithm 1.
**Algorithm 1:** The proposed CMRLCCOA**Input:** Number of coatis (*N*), Number of variables (*D*), and maximum iterations (*M_iter_*).**Output:** Optimal fitness value *f_best_ and X_best_.*1: Construct the initial value for the agents through chaotic maps.2: Computing fitness values for coati populations.3: Using convex lens imaging reverse learning strategy to change the coatis’ position by Equation (11).4: Compare fitness values and retain the optimal fitness values and corresponding positions.5: **While** *t* ≤ *M_iter_*6:    **For** *i* = 1 to *N*/27:      *Igu* = *X_best_*; *I = round*(1 *+ rand*(1,1)).8:      Change the position of the coati by*x_i,j_ = x_i,j_ + b* × (*Igu_j_* − *I*×*x_i,j_*).9:      Update position by Equation (7).10:   **End For**11:   **For** *i* = *N*/2 to *N*12:     *Igu* = *lb* + *rand* × (*ub* − *lb*).13:      **If** fitness(*i*) > fitness(*Igu*)14:        Change the position by *x_i,j_* = *x_i,j_* + *b* × (*Igu_j_* – *I* × *x_i,j_*).15:      **Else**
16:        Change the position by *x_i,j_* = *x_i,j_* + *b* × (*I* × *x_i,j_* − *Igu_j_*).17:      **End If**18:     Update position by Equation (7).19:   **End For**20:   Using Lévy strategy to update the position of the *i*-th coati by Equation (18).21:   Calculate the fitness of coatis.22:   **If** the fitness of coati < fitness(*i*)23:      *x*(*i*) = coati;fit(*i*) = fit(*coati*).24:   **End If**
25:   **For** *i* = 1 to *N*26:      *Lb_Local_* = *lb/t*; *Ub_Local_* = *ub/t.*27:        **If**
*rand* < 0.528:        Update the position of the coatis by*x_i,j_* = *x_i,j_* + (1 − 2*b*) × (*Lb_Local_* + *b* × (*Ub_Local_* − *Lb_Local_*)).29:        Update position by Equation (7).30:      **Else**
31:         **For**
*j* = 1 to *D*32:            *r*_1_ and *r*_2_ is a stochastic number in [0, 1]; *c*_1_ and *c*_2_ is a stochastic number in [−1, 1].33:            Update the position of the leaders using Equations (18) and (19).34:            Calculating acclimatization values of coatis.35:         **End For**36:      **End If**37:   **End For**38:   **For**
*i* = 1 to *N*-139:      **For**
*j* = 1 to *D*40:         Update a uniformly random value *r* in 0 and 1.42:         Update the position of the individuals using Equation (20).43:         Calculating acclimatization values of coatis.44:      **End For**45:   **End For**46:   *t* = *t* + 1,47: **End While**

### 3.6. The Time Complexity of CMRLCCOA

This subsection investigates the time complexity of CMRLCCOA. First, we analyze the COA. The population scale and the amount of problem variables mainly contribute to the time complexity. In the initialization phase, the complexity of COA is O(ND), where *N* is the size of the coati population and *D* is the amount of variables. Among four improvement strategies for the presented CMRLCCOA, the chaotic mapping and lens imaging reverse learning strategies do not increase the complexity. In the first stage, the time complexity is O(NDT) + O(NDT/2) + O(NDT). In the second stage, the complexity is O(NDT) + O((N − 1)DT). Thus, the complexity can be characterized as Equation (22).
*O*(CMRLCCOA) = *O*(Origin coatis) + *O*(Hunting) + *O*(Escaping)= *O*(*ND*(1 + 5/2*T*) + (*N* − 1)*DT*)
(22)



## 4. Numerical Experiments and Comparison with Other Algorithms

In Section 4, we conduct experiments using functions from the CEC2017 and CEC2019 test suites. Among them, it contains 29 functions for CEC2017 and 10 functions for CEC2019. The number of iterations in this experiment is 500 and thirty individuals constitute the entire population. The dimensions of CEC2017 are 50 and 100. The CMRLCCOA is compared to fourteen existing metaheuristic algorithms, including four recognized classical algorithms, PSO (particle swarm optimization) [15], DE (differential evolution) [12], SA (simulated annealing) [16], and ABC (Artificial Bee Colony Algorithm) [61]; six recently proposed algorithms, KOA (Kepler Optimization Algorithm) [62], SWO (Spider Wasp Optimizer) [63], GMO (Geometric Mean Optimizer) [64], OMA (Optical Microscope Algorithm) [65], TROA (Tyrannosaurus Optimization Algorithm) [66], and GO (GOOSE Algorithm) [67]; and three improved algorithms, ISSA (Improved Sparrow Search Algorithm) [68], IGWO (Improved Grey Wolf Optimizer) [69], and EWOA (Enhanced Whale Optimization Algorithm) [70]. Each comparison algorithm is run independently for 20. Finally, the optimum value, the worst value, the mean value, the standard deviation, and the rank for all results are calculated. Furthermore, the Wilcoxon signed rank test is performed to further check the quality of CMRLCCOA. The parameters of the other metaheuristic algorithms are listed in Table 2. Finally, all tests are experimented in Matlab-2020b with a 2.11 GHz quad-core Intel(R) Core(TM) i5 and 8.00 GB.

### 4.1. Introduction to Test Sets

A total of 39 functions were used for testing in this experiment, which come from CEC2017 [71] and CEC2019 [72].

CEC2017 is a test set of intelligent algorithms widely used for a number of optimization problems. The functions are rotated and translated, which in turn increases the difficulty of finding optimization for the algorithms and has a high degree of acceptance. There are four types of benchmark functions: single-peaked, multi-peaked, hybrid, and combined. The single-peak functions (cec01, cec03) are characterized by the fact that there is only a global minimum, not a local minimum. This type of function verifies the convergence of the algorithm. Multi-peak functions (cec04–cec10) have local minima. Such functions verify the competence to get rid of local optima. Algorithms that perform well on these functions generally possess strong exploration capabilities. Hybrid functions (cec11–cec20) have each sub-function assigned a certain weight, which in turn better combines the properties of each sub-function. These functions can effectively verify the ability to find the global optimum. Composite functions (cec21–cec30) have additional bias values and weights for each sub-function. Such functions allow us to assess the accuracy of algorithms. The comprehensive performance will be demonstrated on these functions. To show the details of these functions more clearly, the partial function diagrams are shown in Figure 9.

In addition, this experiment also uses the CEC2019 test set to assess the algorithm’s capability. The CEC2019 test set [73] is a very effective benchmark function set for metaheuristic algorithm performance testing. Among them, cec01–cec03 have different dimensions and ranges, and cannot be moved and rotated. cec04–cec10 are ten-dimensional minimization problems, which can be moved and rotated. This test set is known as the “100-bit challenge” and is often used in international competitions. Some of the functions of CEC 2019 are shown in Figure 10.

### 4.2. Assessment of Indicators

To appraise the efficacy of all algorithms, we take several performance metrics for the analysis, which include the optimal value (Best), the worst value (Worst), the mean value (Ave), the standard deviation (Std), and Rank. Their equations are presented in Equation (23) to Equation (26). The comparison of these metrics can be used to analyze the performance. It is worth noting that Ave can indicate the precision of the algorithm when addressing specific problem categories. The stability can be obtained from Std. Rank is obtained by comparing Ave and Std. If Rank is smaller, it means that the algorithm has a superior performance in solving a particular problem.
(1)Optimum value (Best)
(23)$$\mathrm{Best}=\underset{1\leq i\leq m}{\min}f_{i}^{*}$$
(2)Worst value (Worst)
(24)$$\mathrm{Worst}=\underset{1\leq i\leq m}{\max}f_{i}^{*}$$
(3)Mean value (Ave)
(25)$$\mathrm{Ave}=\frac{1}{m}\sum_{i=1}^{m}f_{i}^{*}$$
(4)Standard deviation (Std)
(26)$$\mathrm{Std}=\sqrt{\frac{1}{m-1}\sum_{i=1}^{m}{\left(f_{i}^{*}-\mathrm{Ave}\right)}^{2}}$$
where *m* corresponds to the tally of independent runs of the algorithms. *f_i_** is the global optimum obtained at the *i*-th independent run.

### 4.3. Effect of Different Chaotic Mapping Functions on CMRLCCOA

Chaotic mapping produces a random sequence for initializing the population, producing a better initial solution [74]. Chaotic mapping produces initial populations in different ways and gives different results. A better strategy produces good initial solutions and is a great enhancement for the subsequent optimization of the algorithm. In this context, 10 functions (taken from CEC2019) are optimized using 10 of the more common chaotic mapping methods to analyze and compare the impact of different chaotic mappings. Table 1 lists these 10 recognized chaotic mapping methods. Table 3 displays the effect and ranking of the 10 chaotic mapping strategies on CMRLCCOA. From the data, it can be seen that the Sine mapping strategy has the best optimization result with the smallest total rank, followed by the Bernoulli mapping strategy and Circle mapping strategy. From the above analysis, it can be concluded that the Sine chaotic mapping strategy performs best in this method and is the first strategy in this method. To show the comparison results more clearly, the experimental data are visualized here, as shown in Figure 11, where the horizontal coordinate is the function of the test set and the vertical coordinate is the algorithm obtained through different mapping methods. And Figure 11 indicates that the Sine chaotic mapping strategy ranks first among the four functions. This also shows that these strategies can be relatively more effective in improving the performance of COA.

### 4.4. Comparison of Optimization Results for CEC2017 

For the purpose of examining CMRLCCOA’s competence in exploring, developing, and jumping out of local optimal solutions, a more competitive test suite, the CEC2017 test suite, was chosen for this paper. CEC2017 [71] is highly recognized and is widely used to validate the performance in all aspects. What is more, this function is not included in this paper because it was not possible to test cec02.

#### 4.4.1. Experimental Results of the CEC2017 Test Suite

In this experiment, CMRLCCOA is run 20 times independently with 14 other algorithms, and finally, Ave, Std, and Rank are calculated. Table A1 and Table A2 show the results obtained by 15 algorithms optimized in 50 and 100 dimensions. The top-ranked values are highlighted in a thick format. 

The observation of the tabular data shows that the CMRLCCOA is ranked first overall with an average ranking of 3 and 2.6552 for *dim* = 50 and 100, respectively; this result shows that the improved CMRLCCOA is better at optimizing in different dimensions and all of them provide excellent output values. Most of the optimal values obtained by CMRLCCOA computation outperform other algorithms. This phenomenon shows that CMRLCCOA is adaptable to different types of functions. CMRLCCOA is able to optimize the nine test functions better in *dim* = 50 (cec03, cec11–12, cec21, cec23, and cec27-30). At *dim* = 100, CMRLCCOA better optimizes 13 test functions (cec03, cec04, cec07, cec11–12, cec14, cec15, cec20–21, cec23–24, cec27–29). GMO also optimizes better and performs better for 10 50-dimensional problems and 100-dimensional problems, second only to CMRLCCOA, and ranked second overall. On the contrary, COA, KOA, SWO, TROA, and GO do not show better optimization ability. In summary, CMRLCCOA significantly outperforms COA as well as the other 13 intelligent optimization algorithms in 50 and 100 dimensions.

The Wilcoxon signed rank test verifies the variability of results obtained by different algorithms [75]. The significance results for *dim* = 50 and 100 are shown in Table A3 and Table A4. “+/=/-” means that the comparative algorithm is significantly better/equal/worse than the CMRLCCOA. The observation of the data reveals that the Wilcoxon test results for COA, KOA, SWO, TROA, and GO are 0/0/29 at *dim* = 50 and 100, indicating that these five algorithms are inferior to CMRLCCOA in all test functions. Meanwhile, the Wilcoxon signed rank results of GMO in two dimensions are 6/10/13 and 6/7/16, which are better. Secondly, ISSA and IGWO also perform better; both of them have six functions better than CMRLCCOA. However, when *dim* = 50 or 100, ABC, EWOA, SA, and DE algorithms perform significantly worse than CMRLCCOA. Therefore, this result also shows that CMRLCCOA can address different types of problems.

Similar to Figure 11, Figure 12 shows two heat maps of the results obtained for all compared algorithms on CEC 2017 for two different dimensions. The vertical coordinates are all algorithms involved in comparison. The performance of all algorithms can be intuitively obtained from the heat map. In the heat map of both dimensions, the corresponding squares of CMRLCCOA proposed in this paper largely show a bluer situation relative to other algorithms. Meanwhile, the GO and TROA rows are always red. This phenomenon indicates that these two algorithms perform poorly on this test set and their performance needs to be improved.

#### 4.4.2. Convergence Curves for Iterations

Figure 13 and Figure 14 show partial convergence curves at *dim* = 50 and 100. From the figure, it can be concluded that CMRLCCOA converges better on cec09, cec20–21, cec24, cec27, and cec28 at *dim* = 50. When *dim* = 100, CMRLCCOA converges better on functions cec07, cec09, cec12, cec20, cec25, and cec27–28. In addition, it can be seen that CMRLCCOA has a larger slope of the curve during the iteration of most functions, which indicates that CMRLCCOA always can converge faster in the early stages. This is made possible by the inclusion of the initialization strategy, which allows the population to explore a larger area. It is able to converge to the neighborhood of the optimum very quickly. In summary, CMRLCCOA can find the optimal solution quickly and can solve some sophisticated optimization questions.

The optimization ability of CMRLCCOA varies when dealing with different functions. As can be noticed from Figure 13 and Figure 14, CMRLCCOA is able to avoid interference factors well in the optimization of both single-peak and multi-peak functions, and both of them converge rapidly to the vicinity of the optimal solution. CMRLCCOA converges faster in hybrid and composite functions. The observation of the curves reveals that the slope of the pre-curve is very large and almost vertical so that the optimal candidate solution can be found in fewer iterations, which indicates that the algorithm has high sensitivity. In addition, it can be found that CMRLCCOA maintains the stability and continuity during the iteration process of most functions, and the convergence accuracy is better. In short, CMRLCCOA is able to solve the functions in CEC2017 efficiently.

#### 4.4.3. Boxplot of Experimental Results

Combined with the convergence curves above, the corresponding box plots are given here. A box-and-line plot is an icon that describes the discrete distribution of data and provides a good description of outliers and skewness in the data. The length of the boxes corresponds to the stability of algorithms. If the box is narrower, the algorithm is more stable and robust. The upper limit of the box-and-line plot is the upper quartile, and the lower line is the lower quartile. Because of the randomized nature of the algorithm, some outliers are generated during the optimization of the problem, and to visually demonstrate the quality of the optimization results, box-and-line plots of the optimization results at *dim* = 50 and 100 are given, as shown in Figure 15 and Figure 16. CMRLCCOA has less variation in the upper and lower distances than the other algorithms, especially at *dim* = 50 for cec22, cec24, cec27, and cec28, and at *dim* = 100 for cec01, cec09, cec11, cec12, ce25, and ce27–28. These functions verify the stability of CMRLCCOA. However, CMRLCCOA also shows some “+” indicators in the box plots of some functions, indicating that the algorithm also produces some outliers with uncertainty and randomness.

In conclusion, for most of the functions, CMRLCCOA is shorter and the upper and lower boundaries are closer together compared to the other 14 algorithms, which indicates that CMRLCCOA is more stable and has better minimum values compared to others.

### 4.5. Comparison of Optimization Results for CEC2019 

This part of the numerical experiments is performed using the 10 functions in CEC2019 [76]. First, the experiment was set up to run 20 independent repetitions. After that, the Ave, Best, Worst, Std, and Rank of the 20 results are computed. Secondly, we obtain the convergence iteration diagrams during the algorithm runs. In addition, all comparison algorithms are consistent with the above experiments. Table 4 illustrates the calculation results. First-ranked data are marked in bold. Finally, box-and-line plots are plotted as a visualization of the quality of the solution results. The radar plot shows more visually how each algorithm ranks on each test function.

#### 4.5.1. Statistical results on CEC2019 

As indicated in Table 4, the average rank of CMRLCCOA is 2.3, which is ranked first overall, better than the others and significantly better than COA. This effect indicates that CMRLCCOA obtains solutions of higher quality relative to the others. In addition, CMRLCCOA significantly optimizes the five functions (cec01, cec04, cec05, cec07, cec08). CMRLCCOA ranks first in terms of computational results in cec01, indicating that it performs well in low-dimensional test functions. CMRLCCOA is superior to the others in cec04 and cec05. This indicates that it is also suitable for higher-dimensional test functions. CMRLCCOA has excellent optimization ability in cec07 and cec08. GMO performs excellently in completing some problems, and has excellent optimization ability, ranking second. GMO outperforms the other algorithms and has a strong optimization ability. In contrast, other algorithms do not solve these functions well.

Table 5 shows the final test results for the Wilcoxon signed rank [77]. A look at the data in Table 5 reveals that the Wilcoxon symbolic rank test outputs for COA, KOA, SWO, GMO, OMA, TROA, GO, PSO, DE, SA, ABC, ISSA, IGWO, and EWOA are 0/1/9, 0/0/10, 0/0/10, 3/2/5, 1/1/8, 0/0/10, 0/0/10, 2/2/6, 2/3/5, 0/3/7, 0/1/9, 2/2/6, 2/1/7, and 0/1/9. It can be found that KOA, SWO, TROA, and GO are not as good as CMRLCCOA on all the tested functions, which can show that CMRLCCOA has better performance and is competitive.

#### 4.5.2. Convergence Curves for Iterations

The convergence curves of comparison algorithms are shown in Figure 17. CMRLCCOA has a very smooth iteration curve and can approach the optimal solution quickly. This shows that CMRLCCOA converges faster than other algorithms, so this algorithm is capable to solve high- and low-dimensional problems. In the experiments on the high-dimensional cec03 function, the CMRLCCOA function converges very fast and moves rapidly to the optimal solution. The CMRLCCOA reaches the neighborhood of the optimum with few iterations during the solution of functions cec05, cec07, and cec09. The convergence is significantly better. The GMO for cec08 is closest to the optimal value and its convergence effect is also excellent. In addition, as shown in Figure 12, the CMRLCCOA algorithm has a very large slope, almost vertical, on the early convergence curves of most functions, indicating that the algorithm has a high sensitivity. Also, PSO, IGWO, OMA, and GMO algorithms show good competence in certain functions. The results show that CMRLCCOA converges faster, gradually approaches the optimal solution, and has better optimization ability than others.

#### 4.5.3. Boxplot of Experimental Results

Figure 18 illustrates a box-and-line plot of CMRLCCOA and other comparative algorithms optimizing the CEC2019 test function. As can be noticed from the figure, it can be noticed that the CMRLCCOA has a lower median case and narrower inter-quartile range, especially in the functions cec01, cec02, cec05, and cec10. It shows that the solutions of CMRLCCOA are more centralized than the other algorithms and are robust. However, CMRLCCOA produces outliers in the optimization process of some functions, such as cec08 and cec09. This phenomenon indicates that this algorithm is unstable to some extent.

#### 4.5.4. Radargram Behavior Analysis

A radar chart is a chart that shows multidimensional data and also shows how much weight is given to each variable in a set of data and can be used to show performance data. To visualize the performance ranking of the different tested functions for all algorithms, Figure 19 illustrates the radar chart of the results for the 10 tested functions sorted. A larger area of the filled portion indicates a lower overall ranking of the algorithm. From Figure 19, it can be concluded that CMRLCCOA has the smallest area, which indicates that CMRLCCOA has the smallest total ranking and the best overall optimization capability. Furthermore, GMO and IGWO also show better performance.

To show the results on the test set more clearly, they are shown here by stacked histograms. As shown in Figure 20, the total height of CMRLCCOA is the lowest. This indicates that CMRLCCOA has relatively the best overall performance and CMRLCCOA is effective. This shows that the mixing of multiple strategies with COA and the construction of CMRLCCOA are effective as well as successful.

## 5. Solutions to Real-World Engineering Optimization Problems

To further test the performance of CMRLCCOA, this section tests it against several excellent metaheuristic algorithms on several complex real-world engineering applications. The comparison algorithms include KOA [62], TROA [66], BWO [29], AO [78], HBA [79], SWO [63], GMO [64], OMA [65], and GO [67]. Their parameters are kept consistent with Table 2. They are not repeated here. The experimental part is as follows.

### 5.1. Single-Stage Cylindrical Gear Reducer (SSCGR)

SSCGR [80] is a kind of reducer that is widely used. The reducer consists of an input shaft and an output shaft. It typically serves as a speed reducer between the primary component and the operational machinery. However, designers tend to focus on the quality and design efficiency of the reducer, thus ignoring the reducer’s consumables, resulting in a huge waste. With the goal of minimizing the number of reducer consumables, SSCGR is established as follows under the premise of meeting the physical model and stability requirements of the reducer: tooth width D, case width B, discrete parameter modulus P and three integer parameters, $\left\{d_{1},d_{2},z_{1}\right\}$, namely. Figure 21 illustrates a diagram of the SSCGR. After specifying the optimization objective, we define the problem as follows:$$X=\left[D,B,P,d_{1}^{\prime },d_{2}^{\prime },z_{1}\right].$$

Minimize
$$\begin{array}{ll} f(X) & =\pi (1.1875x_{1}x_{3}^{2}x_{6}^{2}+0.262x_{1}x_{4}^{2}-0.282x_{1}x_{5}^{2})\\ \; & +21.25x_{1}x_{3}^{2}x_{6}+0.25x_{2}x_{4}^{2}+0.25x_{2}x_{5}^{2}+70x_{4}^{2}\\ \; & +80x_{5}^{2}-21.25x_{1}x_{3}^{2}+0.2x_{1}x_{3}x_{5}x_{6}-0.4x_{1}x_{3}x_{5}) \end{array}$$
subject to
$$g_{1}(X)=\frac{1,367,657.1038}{x_{3}x_{6}\sqrt{x_{1}}}-855\leq 0,\;g_{2}(X)=\frac{6,952,400,000}{-0.0854x_{1}x_{3}^{2}x_{6}^{3}+6.666x_{1}x_{3}^{2}x_{6}^{2}+169x_{1}x_{3}^{2}x_{6}}-261\leq 0,\;g_{3}(X)=\frac{6,952,400,000}{-0.394x_{1}x_{3}^{2}x_{6}^{3}+17.695x_{1}x_{3}^{2}x_{6}^{2}+2824x_{1}x_{3}^{2}x_{6}}-213\leq 0,\;g_{4}(X)=\frac{12.077149x_{2}^{3}}{x_{3}x_{4}^{4}x_{6}}-0.003x_{2}\leq 0,\;g_{5}(X)=\frac{28,456,113.636}{x_{3}x_{4}^{2}x_{6}}\sqrt{1+0.307380\frac{x_{3}^{2}x_{6}^{2}}{x_{2}^{2}}}-55\leq 0,\;g_{6}(X)=\frac{28,456,113.636x_{2}}{x_{3}x_{5}^{3}x_{6}}\sqrt{1+7.684501\frac{x_{3}^{2}x_{6}^{2}}{x_{2}^{2}}}-55\leq 0,\;g_{7}(X)=17-x_{6}\leq 0,\;g_{8}(X)=2-x_{3}\leq 0,\;g_{9}(X)=x_{3}x_{6}-300\leq 0,\;g_{10}(X)=16-x_{1}/x_{3}\leq 0,\;g_{11}(X)=x_{1}/x_{3}-35\leq 0,\;g_{12}(X)=100-x_{4}\leq 0,\;g_{13}(X)=x_{4}-150\leq 0,\;g_{14}(X)=130-x_{5}\leq 0,\;g_{15}(X)-200\leq 0,\;g_{16}(X)=x_{1}+0.5x_{5}+40-x_{2}\leq 0,$$
where the ranges of six design variables being $50\leq x_{1}\leq 150$, $150\leq x_{2}\leq 350$, $0\leq x_{3}\leq 50$, $50\leq x_{4}\leq 150$, $50\leq x_{5}\leq 200$, and $15\leq x_{6}\leq 30$. In addition, *x*_3_ is a discrete variable, the range of the variable is shown in Table 6, and *x*_4_, *x*_5_, and *x*_6_ are integer variables.

Table 7 and Table 8 contain the values of the variables taken and the minimum amount of consumables obtained by CMRLCCOA and other algorithms. It proves that CMRLCCOA outperforms other algorithms on average and is relatively stable. Therefore, CMRLCCOA is preferred for solving this hybrid discrete problem.

### 5.2. Welded Beam Design Problem (WBD)

WBD [81,82] is a classical nonlinear programming problem. Its target is to reduce the production costs associated with the design. Figure 22 illustrates the WDB. This problem is to obtain four constraints that satisfy the constraints of shear stress (*τ*), bending stress (*θ*), bending load (*P*) of the beam bar, end deviation (*δ*), and boundary conditions, such that the cost of fabricating the welded beam is minimized. This question can be explained as follows:

Variant:$$\vec{Q}=[q_{1},q_{2},q_{3},q_{4}]=[h,l,t,b].$$

Minimize
$$f(\vec{Q})=1.10471q_{1}^{2}q_{2}+0.04811q_{3}q_{4}(14.0+q_{2}).$$
subject to
$$f_{1}(\vec{Q})=\tau (\vec{Q})-\tau_{\max}\leq 0,\;f_{2}(\vec{Q})=\sigma (\vec{Y})-\sigma_{\max}\leq 0,\;f_{3}(\vec{Q})=\delta (\vec{Q})-\delta_{\max}\leq 0,\;f_{4}(\vec{Q})=q_{1}-q_{4}\leq 0,\;f_{5}(\vec{Q})=P-Pc(\vec{Q})\leq 0,\;f_{6}(\vec{Q})=0.125-q_{1}\leq 0,\;f_{7}(\vec{Q})=1.10471q_{1}^{2}+0.04811q_{3}q_{4}(14.0+q_{2})-5.0\leq 0.$$
Variable Scope:$$0.1\leq q_{1}\leq 2,\;0.1\leq q_{2}\leq 10,\;0.1\leq q_{3}\leq 10,\;0.1\leq q_{4}\leq 2.$$
where
$$\tau (\vec{Q})=\sqrt{{\left(\tau^{\prime }\right)}^{2}+2\tau^{\prime }\tau^{\prime \prime }\frac{q_{2}}{R}+{\left(\tau^{\prime \prime }\right)}^{2}},\;\tau^{\prime }=\frac{P}{\sqrt{2}q_{1}q_{2}},\;\tau^{\prime \prime }=\frac{MR}{J},\;M=P(L+\frac{q_{2}}{2}),\;R=\sqrt{\frac{q_{2}^{2}}{4}+{\left(\frac{q_{1}+q_{3}}{2}\right)}^{2}},\;J=2\left\{\sqrt{2q_{1}q_{2}}\left[\frac{q_{2}^{2}}{4}+{\left(\frac{q_{1}+q_{3}}{2}\right)}^{2}\right]\right\},\;\sigma (\vec{Q})=\frac{6PL}{q_{4}q_{3}^{2}},\;\delta (\vec{Q})=\frac{6PL^{3}}{Eq_{3}^{2}q_{4}},\;P_{c}(\vec{Q})=\frac{4.013E\sqrt{\frac{q_{3}^{2}q_{4}^{6}}{36}}}{L^{2}}(1-\frac{q_{3}}{L}\sqrt{\frac{E}{64G}}),\;P=6000lb,\;L=14in,\;\delta_{\max}=0.25in,\;G=12\times 10^{6}psi,\;E=30\times 10^{6}psi,\;\tau_{\max}=13,600psi,\;\sigma_{\max}=30,000psi.$$

Table 9 presents the values of the variables and the manufacturing costs. Observing the graphs, it is noticeable that the mean and minimum expenses obtained by CMRLCCOA are less than the comparative algorithms. Therefore, CMRLCCOA can be prioritized when solving similar problems and this algorithm is significantly competitive.

### 5.3. Cantilever Beam Design Problem (CBDP)

This problem is to reduce the weight of the suspension beam arm. The structure of the cantilever beam consists of five hollow cells, each of which has the same cross-sectional thickness [83]. Figure 23 illustrates the structure. The thickness of the crossbar remains fixed and the variables are the widths of the five sections. The issue is presented as follows:

Minimize
$$f(b)=0.0624(b_{1}+b_{2}+b_{3}+b_{4}+b_{5}),\;b_{i}>0.$$
subject to
$$g_{1}(b)=\frac{61}{b_{1}^{3}}+\frac{37}{b_{2}^{3}}+\frac{19}{b_{3}^{3}}+\frac{7}{b_{4}^{3}}+\frac{1}{b_{5}^{3}}\leq 1.$$
Variable range:$$0.01\leq b_{i}\leq 100,\;i=1,2,3,4,5.$$

Table 10 shows the values of the variables obtained from all algorithms, as well as the quality. The best results are marked in bold. Comparing the other results, CMRLCCOA obtains the minimum mass and the results are more stable. This phenomenon demonstrates that CMRLCCOA can solve the cantilever beam problem effectively.

## 6. Real Application: Engineering Optimization Problems

Hypersonic technology is an important milestone in the history of the world’s armaments and equipment, which greatly enriches the content of offensive and defensive confrontation in the adjacent space, represents a country’s ability to develop and utilize space in the future, and is an important symbol of the army’s combat power and survivability, and has a wide range of prospects for application and extremely important military value. The main advantages of hypersonic vehicles are fast flight speed, high flyable altitude from the ground, strong capability of surprise and defense, and great destructive power. In the face of future informationization and intelligent combat, hypersonic vehicles can play a great role by using their characteristics [84].

Since hypersonic vehicles are extremely fast in flight, the environment becomes more complex when the vehicle enters the re-entry or cruise phase, resulting in the need for a control system that is extremely stabilized and at the same time can achieve precise control. Due to the extremely high speed, the missile cannot make a sharp turn in the air. Therefore, in some instances, it is necessary to limit the curvature and turn rate of the aircraft trajectory [85]. Many scholars at home and abroad have also studied this problem. In this section, we will model the path planning of cruise missiles for hypersonic vehicles and apply CMRLCCOA to solve the problem [86].

### 6.1. Background and Establishment of the Model

With the continuous development of weapons technology, the system in the field of military defense and control is being gradually improved. The traditional ballistic missile path may face the risk of being predicted or even intercepted, which is not safe. For the actual ballistic path optimization design problem, different tactical indicators often have different optimization objectives. Hypersonic cruise missiles fly extremely fast and can change their trajectory, thus greatly reducing the risk of interception. These characteristics make it possible to attack targets with very short warning times and at very high speeds. However, current research in this area is relatively small and has not achieved a major breakthrough. In this section, we look at cruise missile trajectories at hypersonic speeds, first considering only the following two conditions:

The hypersonic flight threat area and trajectory map are shown in Figure 24. The constrained region is shown in Figure 25, which shows the positional coordinates of the craft in relation to the radar. In this paper, the radar-centered range of 400 km is used as the solution space, and it is considered that the vertical distance between each defense unit is as far as possible, thus increasing the lateral distance of the interceptor missile.

Hypersonic missile trajectory modeling needs to satisfy certain conditions. Assuming that there are a total of *n* cubic curves, the curvature of the *i*-th curve is denoted as *p_i_*(*t*), 1≤i≤n. The length of the curve is denoted as $l_{i}$, and the derivative of the curvature is denoted as $d_{i}^{\prime }(t)$. The control fixed points are $B_{i,0},B_{i,1},B_{i,2},B_{i,3}$, respectively.

**Optimization Objective**: The length of the missile trajectory curve is the shortest and the curvature derivative is the largest.

**Limitations**: The range of feasible domains, continuity constraints, maximum curvature constraints.

**Decision Variable**: The control vertex.

Minimize
$$k_{1}\sum_{i=1}^{n}l_{i}+k_{2}\sum_{i=1}^{n}D_{i}(t).$$
subject to
$$D_{i}(t)=\min({d^{\prime }}_{i}(t)),\;0\leq d_{i}(t)\leq 1/R,\;i=1,2,\ldots ,n,\;j=0,1,2,3,\;l_{i}=\int_{0}^{1}\sqrt{{\left(p^{\prime }(t)\right)}^{2}+{\left(q^{\prime }(t)\right)}^{2}}dt,\;d_{i}(t)=\frac{p^{\prime }(t)q^{\prime \prime }(t)-q^{\prime }(t)p^{\prime \prime }(t)}{{\left({\left(p^{\prime }(t)\right)}^{2}+{\left(q^{\prime }(t)\right)}^{2}\right)}^{3/2}}.$$
where *k*_1_ and *k*_2_ are the weighting factor.

The above is a more complex minimization problem proposed in this paper. Next, CMRLCCOA is used to solve it for hypersonic missiles.

### 6.2. Solving the Model

CMRLCCOA, as well as KOA, TROA, BWO, AO, HBA, SWO, GMO, OMA, and GO, is applied to the hypersonic cruise ballistic optimization problem. Table 11 lists the optimal ballistic lengths solved by the 10 algorithms. Observing the table, it can be observed that the shortest length obtained by CMRLCCOA is 51.231801 km. This result is less than the results calculated by the others. This phenomenon indicates that CMRLCCOA performs better.

## 7. Summary and Outlook

In this paper, four strategies are used to improve COA, which leads to the proposed CMRLCCOA. First, in the initialization population phase, the coati population is initialized using the Sine chaotic mapping function to avoid population randomization. Second, a lens imaging reverse learning strategy is applied to renew the location of the coati population again. This strategy can expand the search space and enhance the quality of coati populations. Then, the Lévy flight strategy allows coatis to move over a wide range in the search space, reducing the iguana constraint. This method makes the algorithm better at finding a global optimum. Finally, the use of the crossover strategy reduces the search blind spots and improves the algorithm’s accuracy. Experiments are conducted in CEC2017 and CEC2019 test suites, where 50 and 100 dimensions are used in CEC2017. Lastly, the optimization results derived from CMRLCCOA are compared with COA, six new algorithms proposed in the last two years, four classical and well-recognized algorithms, and three enhanced algorithms. Then, we find that the newly proposed CMRLCCOA has better results and higher performance. In addition, CMRLCCOA is able to optimize to obtain better solutions to the three engineering problems. Finally, this paper also establishes a model of a hypersonic vehicle cruise ballistic problem. CMRLCCOA performs best in solving the hypersonic cruise ballistic trajectory optimization, reflecting the strong optimization capability and stability of CMRLCCOA.

To conclude, this study has strong scientific and practical value. The possible future work is as follows: Although the proposed CMRLCCOA has enhanced optimization capability and accelerated convergence speed, it still has areas of improvement in terms of computational complexity and computation time. CMRLCCOA will be further optimized for this problem in the follow-up work. In addition, we will continue to study on the basis of CMRLCCOA to obtain better solutions and apply it to address many complicated optimization problems, including route planning [87,88], image division problems [89], workshop scheduling [90], feature selection [91,92], shape optimization [76], and engineering optimization [93], and further expand the application field of intelligent algorithms.

## Figures and Tables

**Figure 1 biomimetics-09-00399-f001:**
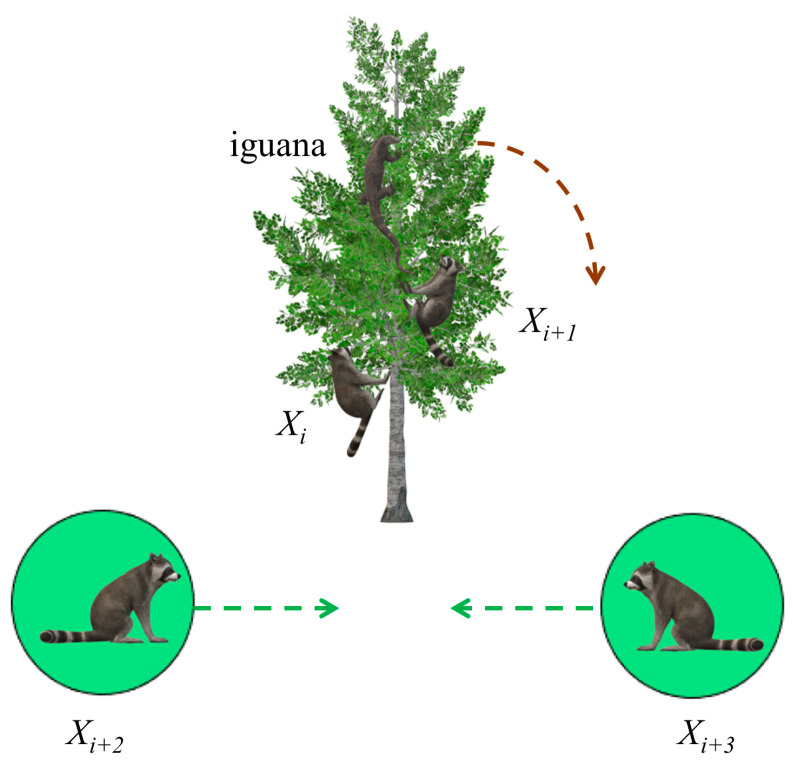
Coatis attacking iguana.

**Figure 2 biomimetics-09-00399-f002:**
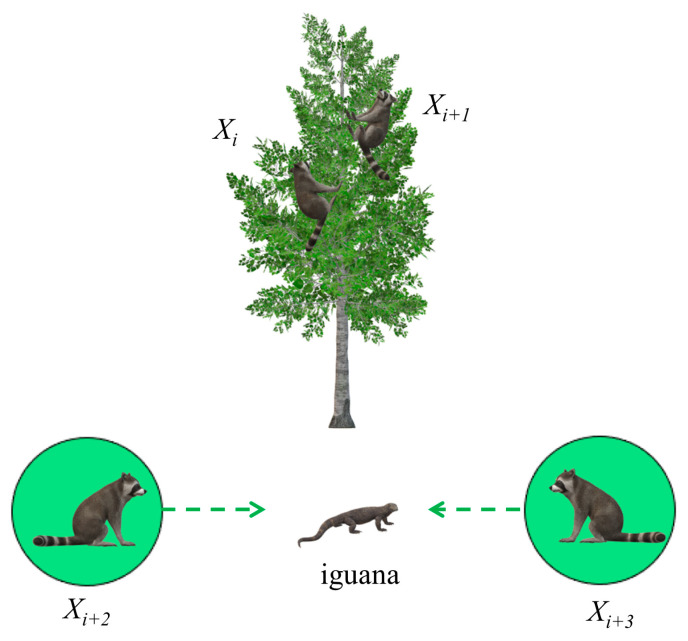
Iguana update location.

**Figure 3 biomimetics-09-00399-f003:**
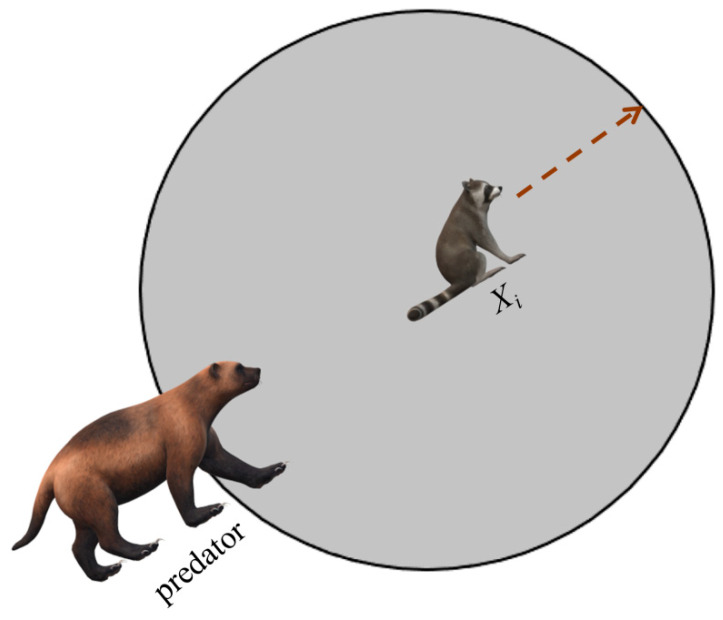
A schematic illustration of the behavior of a coati fleeing from a predator.

**Figure 4 biomimetics-09-00399-f004:**
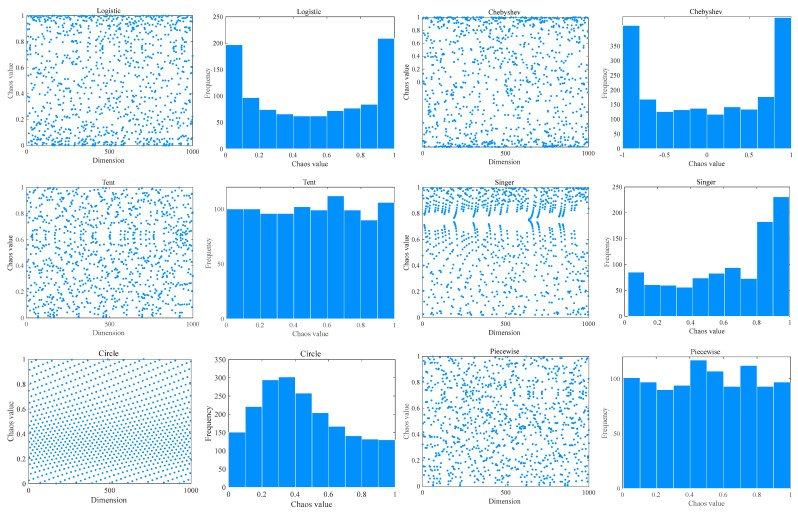
The schematics of chaotic mapping functions.

**Figure 5 biomimetics-09-00399-f005:**
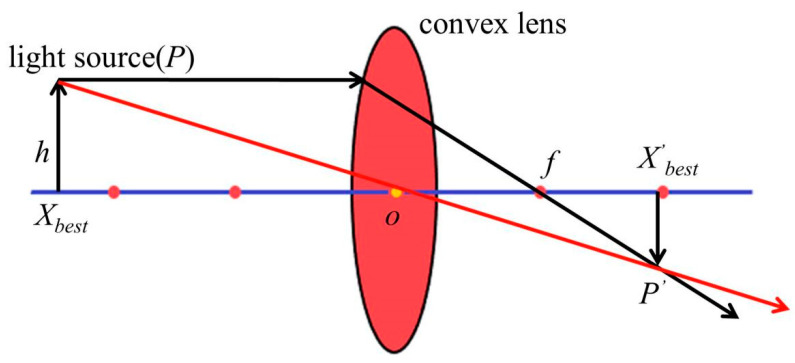
A schematic diagram of the principle of convex lens imaging.

**Figure 6 biomimetics-09-00399-f006:**
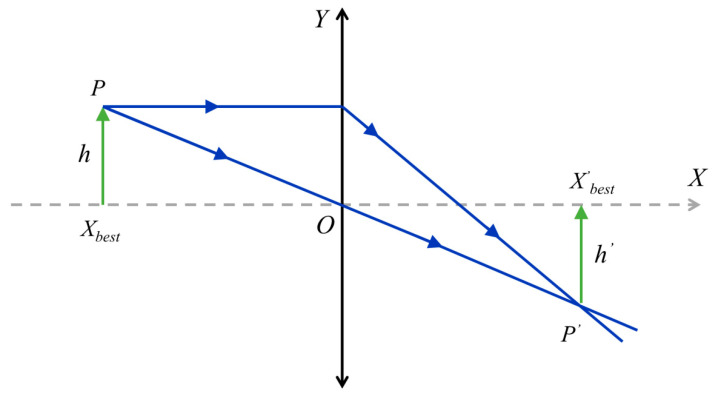
Schematic diagram of reverse learning strategy for convex lens imaging.

**Figure 7 biomimetics-09-00399-f007:**
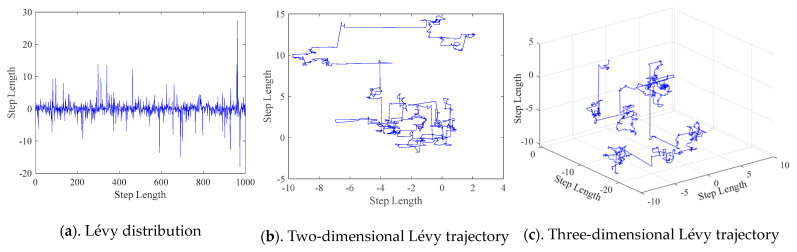
The Lévy flight trajectory.

**Figure 8 biomimetics-09-00399-f008:**
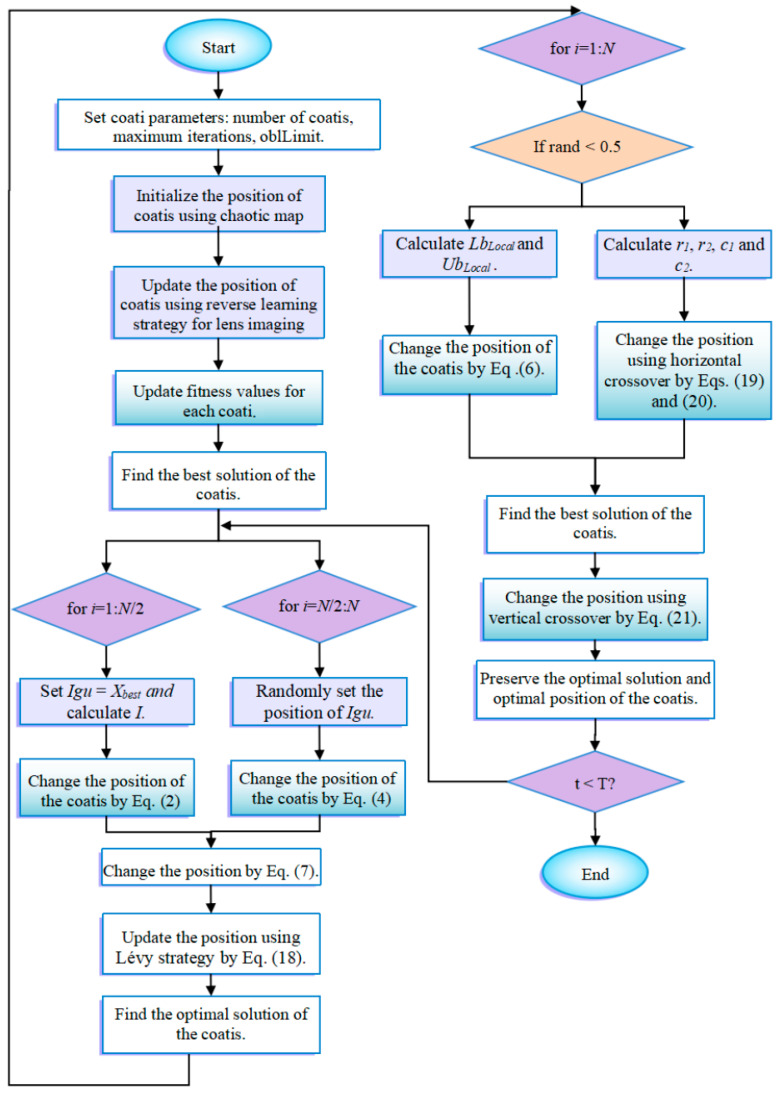
Flowchart of CMRLCCOA.

**Figure 9 biomimetics-09-00399-f009:**
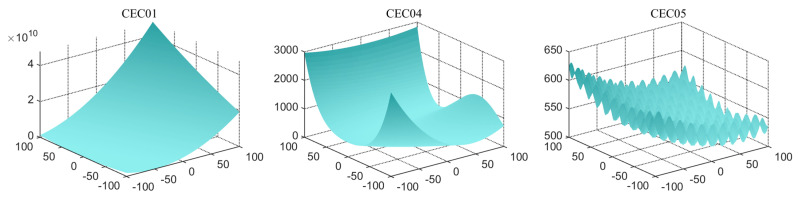
The partial function diagrams in CEC 2017.

**Figure 10 biomimetics-09-00399-f010:**
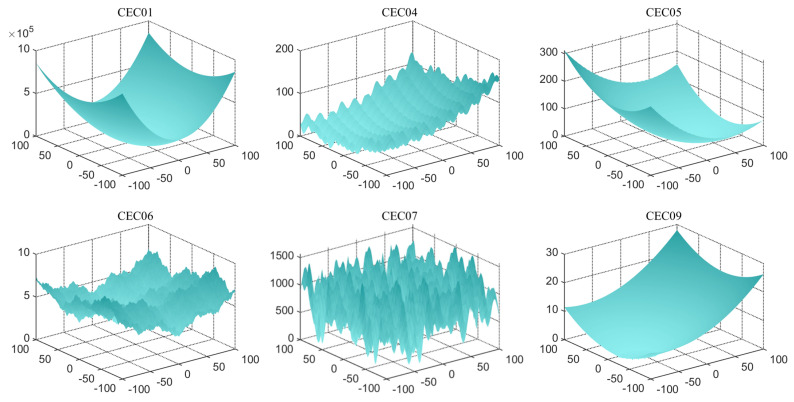
The partial function diagrams in CEC 2019.

**Figure 11 biomimetics-09-00399-f011:**
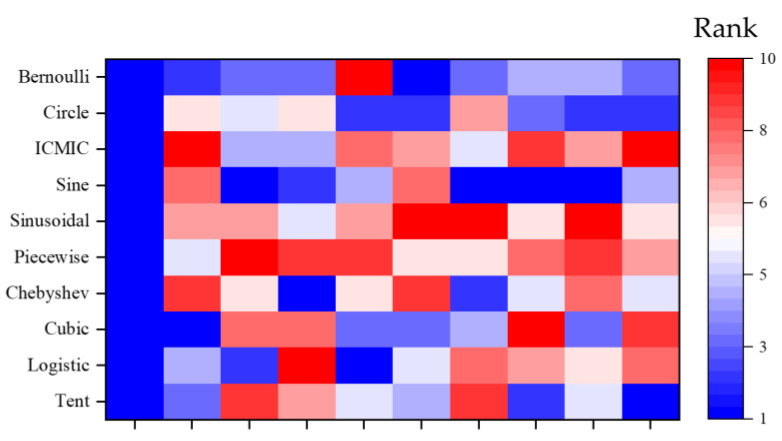
Heat maps corresponding to the rankings obtained from different chaotic mappings.

**Figure 12 biomimetics-09-00399-f012:**
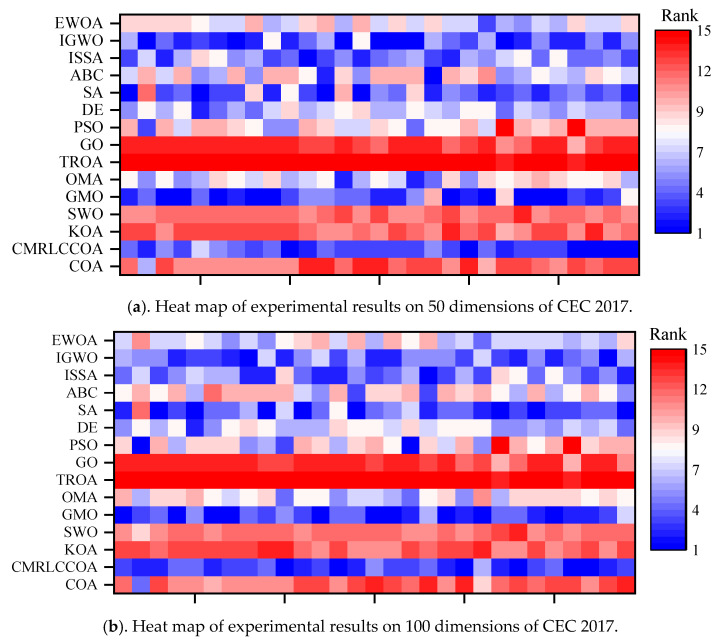
Heat map of experimental results on CEC 2017.

**Figure 13 biomimetics-09-00399-f013:**
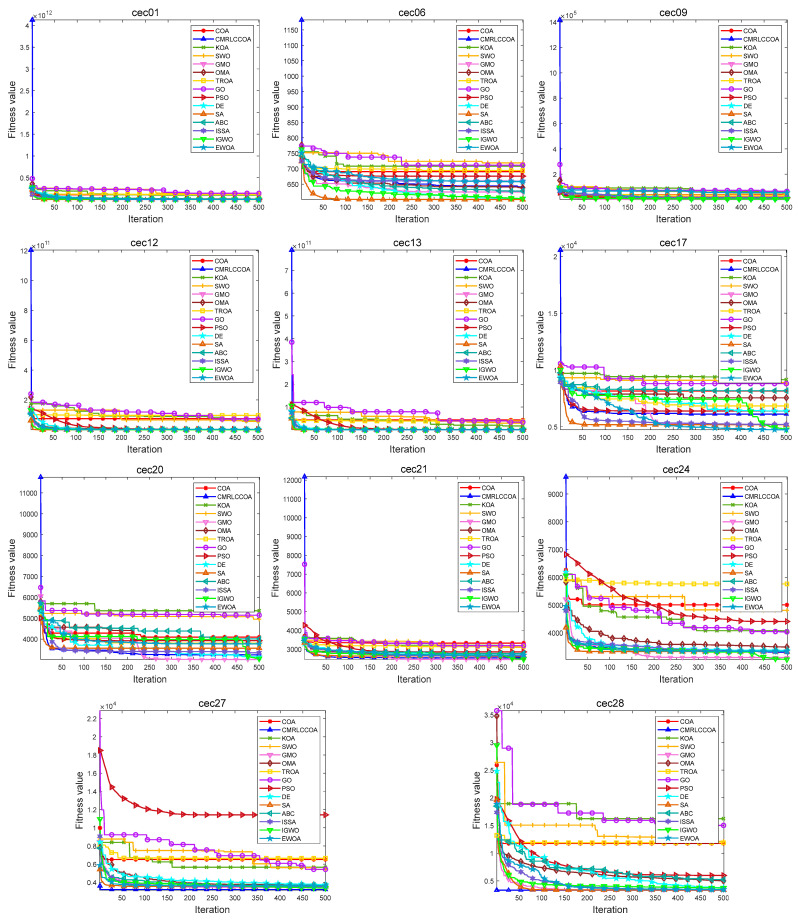
The iteration profile of CMRLCCOA with other comparative algorithms for *dim* = 50.

**Figure 14 biomimetics-09-00399-f014:**
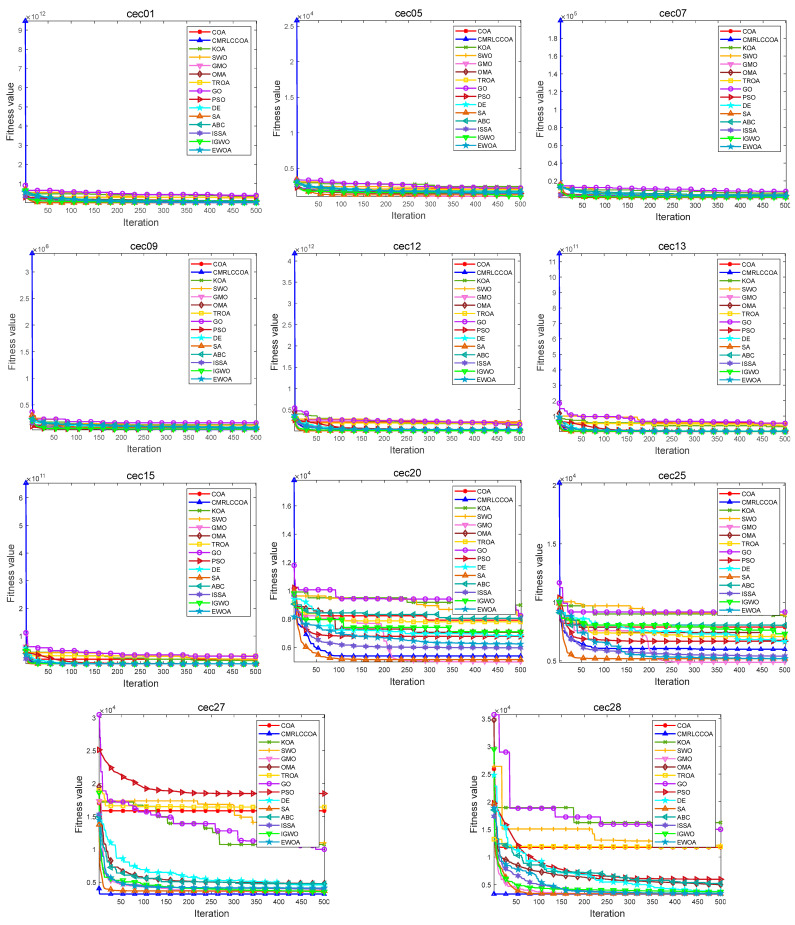
The iteration profile of CMRLCCOA with other comparative algorithms for *dim* = 100.

**Figure 15 biomimetics-09-00399-f015:**
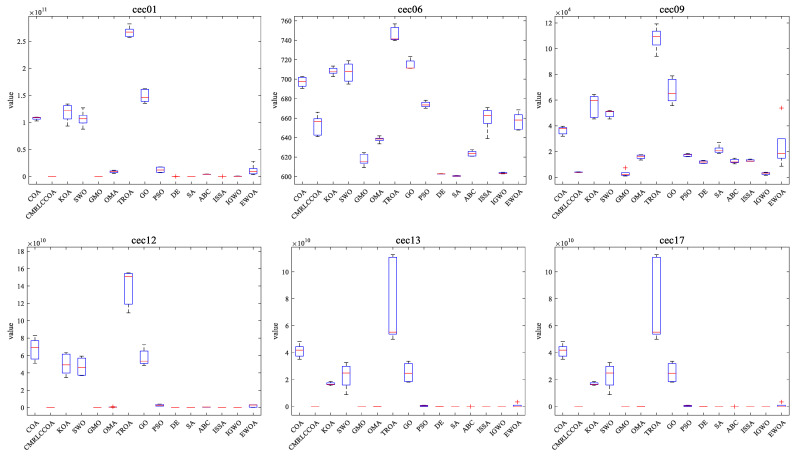
Boxplot of CMRLCCOA algorithm with other comparative algorithms for *dim* = 50.

**Figure 16 biomimetics-09-00399-f016:**
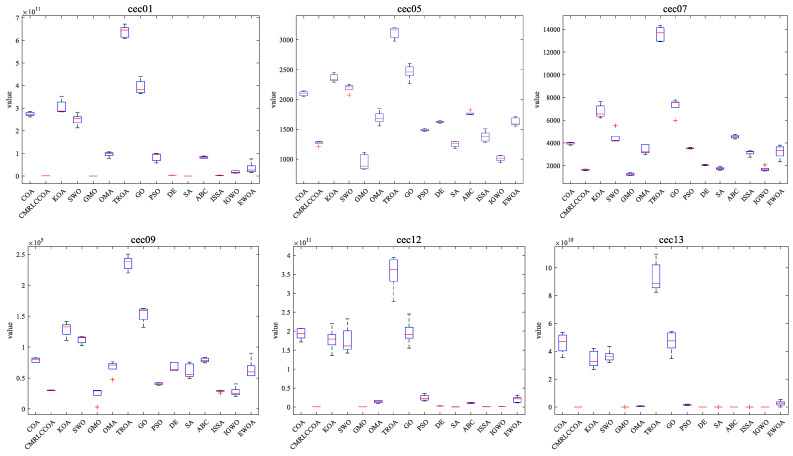
Boxplot of CMRLCCOA algorithm with other comparative algorithms for *dim* = 100.

**Figure 17 biomimetics-09-00399-f017:**
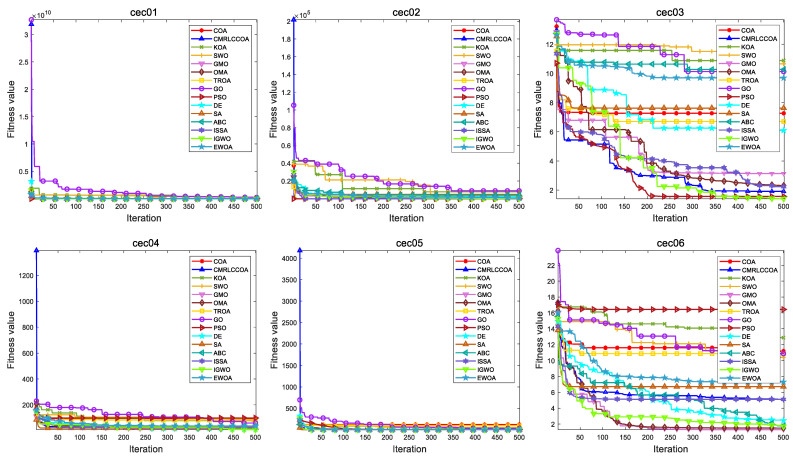
The iteration profile of CMRLCCOA with other comparative algorithms in CEC2019.

**Figure 18 biomimetics-09-00399-f018:**
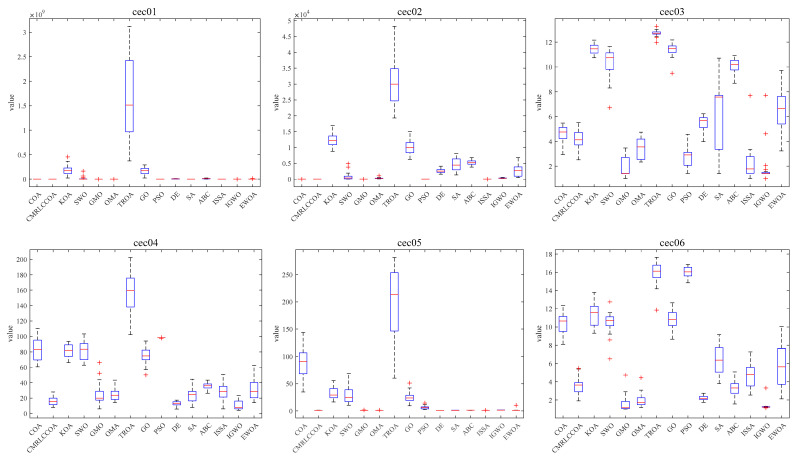
Boxplot of CMRLCCOA algorithm with other comparative algorithms in CEC2019.

**Figure 19 biomimetics-09-00399-f019:**
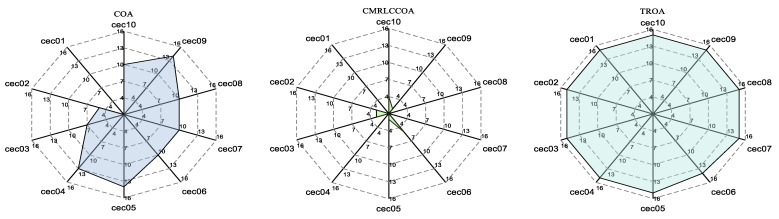
Radar chart of CMRLCCOA and other algorithms on CEC2019.

**Figure 20 biomimetics-09-00399-f020:**
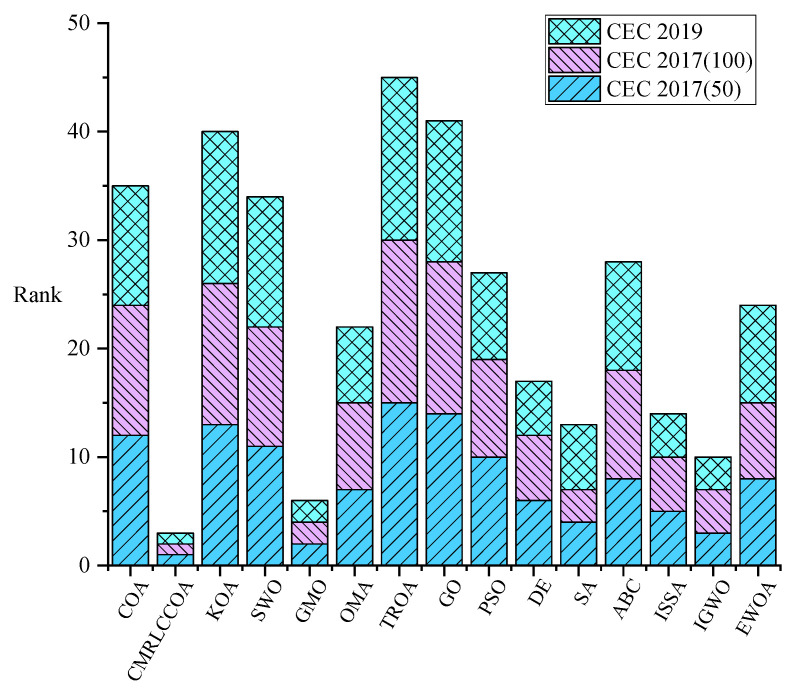
The stacked histogram on all test sets.

**Figure 21 biomimetics-09-00399-f021:**
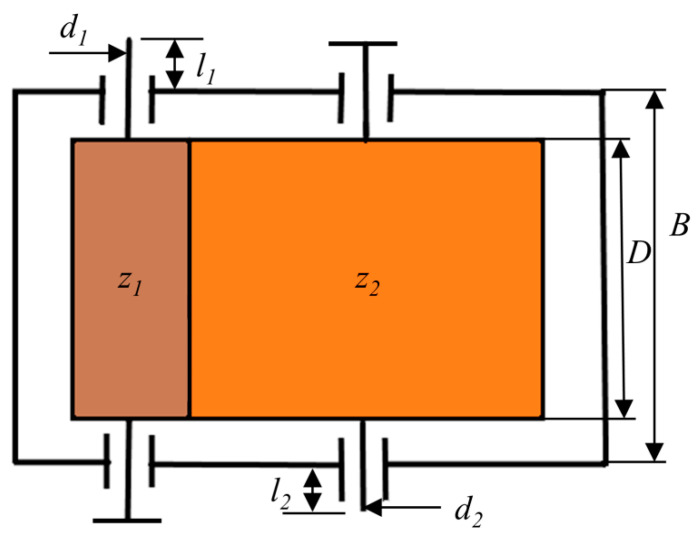
Plane schematic diagram of SSCGR.

**Figure 22 biomimetics-09-00399-f022:**
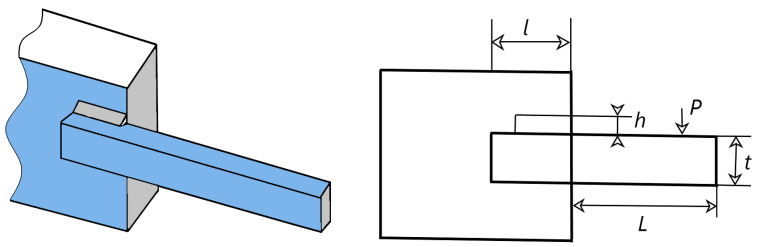
Schematic of WBD.

**Figure 23 biomimetics-09-00399-f023:**
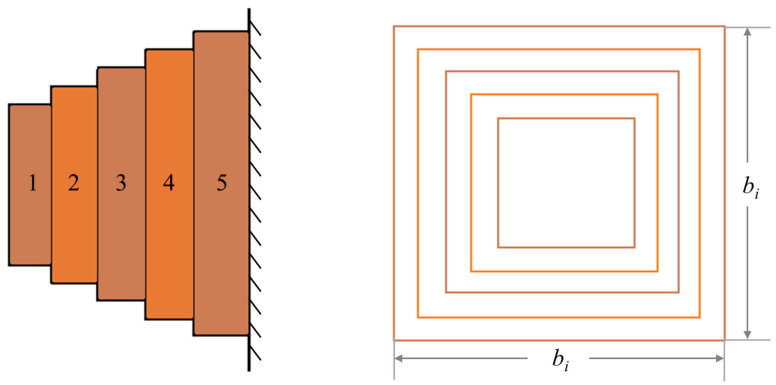
Illustration of CBDP.

**Figure 24 biomimetics-09-00399-f024:**
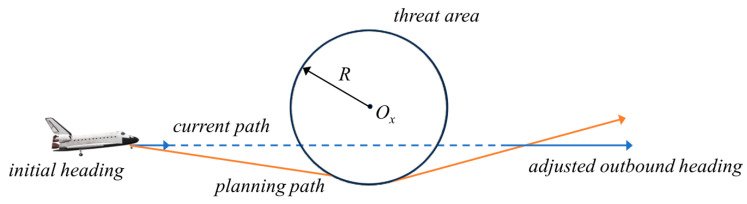
Threat area and path planning map.

**Figure 25 biomimetics-09-00399-f025:**
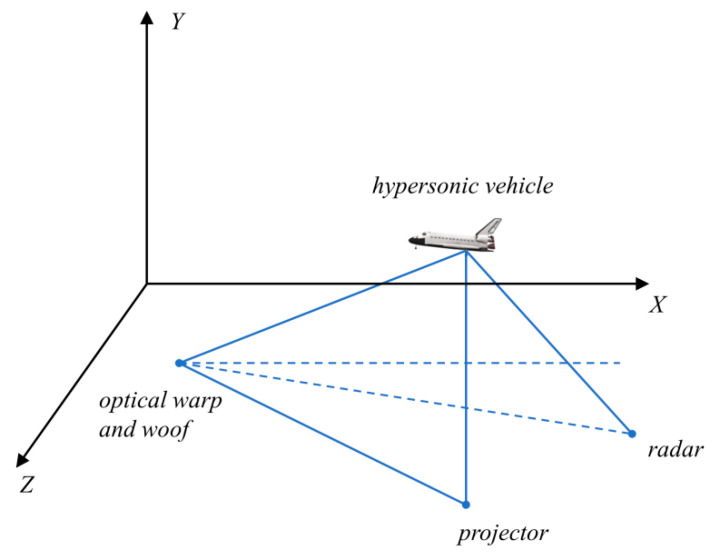
Vehicle versus radar position plot.

**Table 1 biomimetics-09-00399-t001:** Introduction to ten common chaotic mapping functions.

NO.	Function Name	Function Definition	Parameter
1	Sinusoidal [48]	$z_{k+1}=bz_{k}^{2}\sin(\pi \;z_{k})$	*b* = 2.3 and *z*_0_ = 0.7.
2	Logistic [49]	$z_{k+1}=bz_{k}(1-z_{k})$	*z_k_* is the *k*th chaotic number, and *z*_0_∈(0, 1).
3	Tent [50]	$z_{k+1}=\left\{\begin{array}{l} \frac{z_{k}}{\alpha },z_{k}\in (0,\alpha ]\\ \frac{1-z_{k}}{1-\alpha },z_{k}\in (z_{k},1] \end{array}\right.$	-
4	Gauss/Mouse [51]	$z_{k+1}=\left\{\begin{array}{l} 0,z_{k}=0\\ \mathrm{mod}(\frac{\mu }{z_{k}},1),otherwise \end{array}\right.$	Generates chaotic sequences in (0, 1).
5	Circle [52]	$z_{k+1}=z_{k}+b-\mathrm{mod}(\frac{a}{2\pi }\sin(2\pi z_{k}),1)$	*a =* 2.2 and *b =* 0.5.
6	Chebyshev [53]	$z_{k+1}=\cos(k\cos^{-1}(z_{k}))$	-
7	Singer [54]	$z_{k+1}=\mu (7.86z_{k}-23.31z_{k}^{2}+28.75z_{k}^{3}-13.30z_{k}^{4})$	*µ* is set between 0.9 and 1.08.
8	Bernoulli [55]	$z_{k+1}=\left\{\begin{array}{l} \frac{z_{k}}{1-\lambda },z_{k}\in (0,1-\lambda ]\\ \frac{z_{k}-1+\lambda }{\lambda },z_{k}\in (1-\lambda ,1) \end{array}\right.$	-
9	ICMIC [56]	$z_{k+1}=\sin\left(\frac{a}{z_{k}}\right)$	*a*∈(0, ∞).
10	Piecewise [57]	$x_{k+1}=\left\{\begin{array}{l} z_{k}/q,z_{k}\in [0,q)\\ \left(z_{k}-q)/(0.5-q),z_{k}\in [q,\;0.5\right)\\ \left(1-n-z_{k})/(0.5-n),z_{k}\in [0.5,\;1-n\right)\\ \left(1-z_{k})/n,z_{k}\in [1-n,\;1\right) \end{array}\right.$	*n*∈(0, 0.5) and *z_k_*∈[0, 1].

**Table 2 biomimetics-09-00399-t002:** Algorithm-related information.

Algorithm	Year	Parameter Name	Value
PSO	1995	Inertia weight	Decreasing linearly from 0.9 to 0.1
		Velocity range	0.1 times the size of the variable
		Cognitive and social factors	*c*1 = 2, *c*2 = 2
DE	1995	Scaling factor	0.5
		Crossover probability	0.5
SA	1953	-	-
ABC	2005	Limit	20
KOA	2023	Velocity	*Tc* = 3, *M*_0_ = 0.1, λ = 15
SWO	2023	Hunting and nesting weight	0.5
GMO	2023	Dual-fitness index	*α =* 0.05*, Pa_2_ =* 0.2*, Prb =* 0.2
OMA	2023	Space	0.55
TROA	2023	Hunting success rate	[0.1,.1]
GO	2024	Stone weight	*p*1 = 5, *p*2 = 0.001, *p*3 = 0.3
IGWO	2021	*α*	Decreases linearly from 2 to 0
EWOA	2023	a	Decreased from 2 to 0
		b	2

**Table 3 biomimetics-09-00399-t003:** Results of chaotic mapping functions in the CEC2019 test set.

Function	Chaos Mapping, Rank									
	Tent		Logistic		Cubic		Chebyshev		Piecewise	
CEC01	1	1	1	1	1	1	1	1	1	1
CEC02	4.8343428	3	4.9086048	4	4.7536217	1	4.9480054	9	4.9127265	5
CEC03	4.5113105	9	3.6855279	2	4.4167636	8	4.0965744	6	4.5839437	10
CEC04	21.433848	7	30.186825	10	24.332956	8	16.31391	1	27.812378	9
CEC05	1.1617451	5	1.1431999	1	1.1515950	3	1.1631781	6	1.194418	9
CEC06	3.3690697	4	3.3841554	5	3.2145844	3	3.6269374	9	3.4008807	6
CEC07	732.95347	9	725.80671	8	614.88261	4	584.74687	2	618.38094	6
CEC08	2721.4771	2	2647.8950	7	2851.1804	10	2604.0760	5	2792.7936	8
CEC09	3.6744951	5	3.8464906	6	3.9101588	3	3.7696891	8	3.8590904	9
CEC10	1.1916796	1	1.2024768	8	1.1872587	9	1.2116937	5	1.2209108	7
Ave rank	4.6		5.2		5		5.2		7	
Final rank	4		6		5		6		10	
Function	Chaos mapping, Rank									
	Sinusoidal		Sine		ICMIC		Circle		Bernoulli	
CEC01	1	1	1	1	1	1	1	1	1	1
CEC02	4.9393187	7	4.9476647	8	4.9675991	10	4.9228798	6	4.7674341	2
CEC03	4.1652904	7	3.6007047	1	3.8968772	4	3.9812347	5	3.7443864	3
CEC04	19.948643	5	16.530138	2	19.427083	4	20.471053	6	18.128766	3
CEC05	1.1669640	7	1.1614536	4	1.1923656	8	1.1509138	2	1.200149	10
CEC06	3.6426736	10	3.4905846	8	3.4168389	7	3.1405742	2	2.8830131	1
CEC07	813.11549	10	580.91894	1	615.26845	5	688.45524	7	604.38442	3
CEC08	2734.7201	6	2530.4876	1	2646.3779	9	2686.5244	3	2650.1646	4
CEC09	3.7708771	10	3.5869369	1	3.8648624	7	3.7574923	2	3.7603243	4
CEC10	1.2469245	6	1.1435522	4	1.2066848	10	1.1592935	2	1.1908522	3
Total rank	6.9		3.1		6.5		3.6		3.4	
Final rank	9		1		8		3		2	

**Table 4 biomimetics-09-00399-t004:** Comparison results of CMRLCCOA and different algorithms in CEC2019.

Function	Index	COA	CMRLCCOA	KOA	SWO	GMO	OMA	TROA	GO	PSO	DE	SA	ABC	ISSA	IGWO	EWOA
cec01	Ave	1	1	1.84E+08	1.48E+07	1	2.84E+05	1.67E+09	1.69E+08	1	5.20E+06	1	1.05E+07	1	5.79E+04	2.37E+06
	Worst	1	1	4.54E+08	1.64E+08	1	1122748	3.12E+09	2.92E+08	1	11058720	1	23234725	1	314276	13879628
	Best	1	1	2.44E+07	9.39E+01	1	3.36E+04	3.74E+08	2.25E+07	1	5.33E+05	1	2.90E+06	1	1.23E+01	5.95E+04
	Std	0	0	9.53E+07	3.73E+07	9.92E-08	2.48E+05	8.49E+08	7.31E+07	0	3.20E+06	0	5.06E+06	0	8.21E+04	3.05E+06
	Rank	1	1	14	12	6	8	15	13	1	10	1	11	1	7	9
cec02	Ave	5	4.818	12317.590	861.263	4.901	294.950	30233.500	9853.688	4.524	2666.823	4679.524	5229.067	4.254	334.713	2764.978
	Worst	5	5.000	16901.760	4888.710	5.016	1082.046	48225.370	15077.270	4.767	4079.375	8195.320	6785.122	4.374	520.294	6792.063
	Best	5	4.227	8797.222	6.154	4.484	103.766	19285.510	6266.771	4.313	1603.955	1354.395	3788.566	4.217	184.893	494.261
	Std	1.04E-06	0.262	1984.790	1278.936	0.162	201.267	7041.787	2262.994	0.150	716.899	2066.515	747.501	0.034	88.850	1925.375
	Rank	5	3	14	8	4	6	15	13	2	9	11	12	1	7	10
cec03	Ave	4.622	2.106	11.433	10.332	1.993	3.473	12.699	11.374	2.815	5.432	6.197	10.087	2.278	1.864	6.742
	Worst	5.475	5.512	12.168	11.634	3.475	4.739	13.278	12.170	4.564	6.231	10.712	10.935	7.686	7.712	9.710
	Best	2.943	1.317	10.743	6.714	1.000	2.343	11.973	9.492	1.413	3.979	1.409	8.684	1.002	1.001	3.237
	Std	0.661	0.809	0.384	1.179	0.833	0.823	0.255	0.544	0.879	0.675	2.878	0.546	1.420	1.535	1.788
	Rank	7	3	14	12	2	6	15	13	5	8	9	11	4	1	10
cec04	Ave	83.126	11.211	81.242	81.491	25.163	24.406	155.292	74.851	98.306	13.142	23.918	36.051	28.804	12.690	30.922
	Worst	110.190	27.918	93.461	103.081	66.180	43.391	202.658	93.995	98.506	17.302	44.599	43.256	50.679	23.203	62.484
	Best	60.535	3.992	65.997	62.612	5.975	14.378	102.226	50.088	97.510	5.662	7.965	26.430	5.975	4.008	13.934
	Std	14.114	5.133	8.691	10.972	14.740	7.680	26.491	10.343	0.398	2.749	9.215	3.976	10.740	6.509	12.803
	Rank	13	1	11	12	6	5	15	10	14	3	4	9	7	2	8
cec05	Ave	88.662	1.209	32.158	28.849	1.226	1.232	201.010	25.001	6.406	1.349	1.213	1.485	1.138	1.606	2.084
	Worst	143.853	1.345	55.645	68.663	1.885	1.521	281.706	51.233	14.646	1.421	1.608	1.637	1.426	1.793	10.596
	Best	34.877	1.046	16.528	10.402	1.039	1.041	60.285	9.520	2.341	1.062	1.012	1.306	1.034	1.423	1.030
	Std	29.671	0.103	10.825	15.632	0.226	0.113	64.150	9.822	3.244	0.042	0.165	0.085	0.093	0.093	2.796
	Rank	14	2	13	12	4	5	15	11	10	6	3	7	1	8	9
cec06	Ave	10.416	3.083	11.411	10.418	1.612	1.980	15.880	10.706	16.019	2.226	6.393	3.298	4.713	1.838	5.789
	Worst	12.364	5.473	13.788	12.757	4.735	4.450	17.642	12.658	16.858	2.752	9.150	5.090	7.272	3.322	10.030
	Best	8.101	1.915	9.318	6.511	1.013	1.162	11.850	8.652	14.849	1.745	3.805	1.568	2.555	1.138	2.122
	Std	1.128	0.973	1.219	1.247	0.915	0.751	1.312	1.115	0.586	0.275	1.533	0.863	1.360	0.457	2.247
	Rank	10	5	13	11	1	3	14	12	15	4	9	6	7	2	8
cec07	Ave	1888.543	630.452	2214.700	1984.343	1127.429	1488.903	2920.239	2057.648	1966.054	692.845	720.485	1429.335	1002.165	695.429	966.935
	Worst	2213.910	998.426	2555.673	2365.470	2086.663	1744.456	3437.083	2347.304	2156.438	919.701	1130.655	1711.483	1582.198	1431.710	1526.280
	Best	1364.789	271.372	1825.462	1408.175	357.282	1256.515	2219.054	1619.865	1818.921	411.468	245.729	1155.446	519.223	5.542	125.939
	Std	238.431	203.125	183.251	250.080	521.197	144.181	328.974	210.137	111.793	132.633	229.616	164.750	253.134	473.452	318.565
	Rank	10	1	14	12	7	9	15	13	11	2	4	8	6	3	5
cec08	Ave	4.839	3.174	5.202	5.084	3.991	4.055	5.601	5.110	5.517	3.794	4.279	4.478	4.293	3.235	4.418
	Worst	5.055	4.638	5.476	5.352	4.931	4.333	5.874	5.475	5.560	4.034	5.203	4.777	5.000	3.865	4.932
	Best	4.149	3.275	4.997	4.816	3.133	3.342	5.147	4.822	5.493	3.371	3.502	4.104	3.455	2.059	3.825
	Std	0.227	0.368	0.140	0.147	0.510	0.239	0.179	0.168	0.022	0.198	0.434	0.177	0.398	0.562	0.293
	Rank	10	1	13	11	4	5	15	12	14	3	6	9	7	2	8
cec09	Ave	3.726	1.209	2.528	2.312	1.095	1.495	6.666	2.264	1.234	1.275	1.486	1.277	1.330	1.216	1.435
	Worst	4.445	1.385	3.225	3.296	1.208	1.829	8.271	3.043	1.377	1.337	1.904	1.362	1.475	1.297	1.723
	Best	2.779	1.056	1.925	1.544	1.045	1.251	5.311	1.601	1.083	1.185	1.058	1.169	1.144	1.081	1.196
	Std	0.488	0.084	0.313	0.453	0.038	0.146	0.733	0.401	0.098	0.037	0.225	0.056	0.108	0.047	0.156
	Rank	14	2	13	12	1	10	15	11	4	5	9	6	7	3	8
cec10	Ave	21.455	21.017	21.788	21.635	19.365	21.381	22.101	21.737	20.985	20.934	21.019	21.452	21.368	21.434	21.462
	Worst	21.624	21.464	21.992	21.832	21.601	21.566	22.367	21.978	20.987	21.191	21.115	21.620	21.592	21.577	21.628
	Best	21.205	21.145	21.413	21.373	1.001	20.591	21.709	21.464	20.983	18.057	21.000	21.253	21.027	21.263	21.185
	Std	0.121	0.082	0.140	0.143	6.123	0.201	0.193	0.126	0.001	0.705	0.032	0.098	0.162	0.088	0.106
	Rank	10	4	14	12	1	7	15	13	3	2	5	9	6	8	11
Mean Rank		9.4	2.3	13.3	11.4	3.6	6.4	14.9	12.1	7.9	5.2	6.1	8.8	4.7	4.3	8.6
Final Ranking		11	1	14	12	2	7	15	13	8	5	6	10	4	3	9

**Table 5 biomimetics-09-00399-t005:** Significance of CMRLCCOA and different algorithms in CEC2019.

Function	COA	KOA	SWO	GMO	OMA	TROA	GO	PSO	DE	SA	ABC	ISSA	IGWO	EWOA
cec01	NaN/=	8.01E-09/-	6.08E-08/-	2.99E-08/-	8.01E-09/-	8.01E-09/-	6.08E-01/-	NaN/=	6.51E-08/-	NaN/=	8.01E-09/-	NaN/=	8.01E-09/-	7.93E-09/-
cec02	1.57E-08/-	6.80E-08/-	9.23E-06/-	1.33E-02/-	6.80E-08/-	7.33E-07/-	6.80E-08/-	7.58E-04/+	9.21E-08/-	6.33E-08/-	3.76E-06/-	7.95E-07/+	6.31E-08/-	6.80E-08/-
cec03	3.85E-02/-	7.99E-08/-	6.80E-08/-	7.95E-07/+	1.93E-02/-	6.80E-08/-	6.31E-07/-	2.22E-04/-	1.81E-05/-	1.79E-02/-	6.80E-08/-	7.58E-06/-	6.67E-06/+	9.75E-06/-
cec04	6.80E-08/-	6.04E-09/-	3.88E-09/-	4.39E-02/-	1.01E-02/-	9.38E-06/-	9.03E-06/-	6.80E-08/-	8.59E-02/=	9.05E-03/-	7.92E-09/-	1.79E-04/-	6.04E-03/-	1.61E-04/-
cec05	7.32E-07/-	6.80E-08/-	7.90E-08/-	3.10E-01/=	5.08E-01/=	3.20E-05/-	5.30E-06/-	6.80E-08/-	1.33E-01/=	6.95E-01/=	5.23E-07/-	3.37E-02/+	9.11E-09/-	2.62E-01/=
cec06	6.80E-08/-	4.26E-07/-	6.80E-08/-	3.99E-06/+	1.25E-05/+	6.80E-08/-	7.69E-07/-	9.60E-04/-	1.81E-05/+	2.32E-05/-	4.41E-01/-	1.79E-02/-	2.56E-07/+	2.561E-03/-
cec07	3.68E-08/-	7.30E-08/-	6.80E-08/-	3.97E-03/-	6.80E-08/-	6.80E-08/-	9.01E-06/-	7.31E-06/-	2.50E-01/=	2.62E-01/=	2.50E-07/-	8.29E-05/-	8.18E-01/=	3.75E-04/-
cec08	7.88E-05/-	6.80E-08/-	9.10E-06/-	3.99E-06/-	1.25E-05/-	6.36E-09/-	6.80E-08/-	6.80E-08/-	1.81E-05/-	2.36E-06/-	4.41E-01/=	1.79E-02/-	2.56E-07/-	2.56E-03/-
cec09	8.21E-08/-	6.80E-08/-	6.80E-08/-	1.81E-05/+	6.01E-07/-	6.80E-08/-	3.67E-04/-	2.98E-01/=	2.80E-03/-	1.61E-04/-	6.04E-03/-	9.21E-04/-	2.39E-02/-	1.41E-05/-
cec10	9.05E-03/-	1.23E-07/-	1.38E-06/-	1.26E-01/=	7.64E-02/-	6.80E-08/-	6.80E-08/-	5.33E-05/+	2.56E-07/+	6.80E-08/-	3.06E-03/-	5.08E-01/=	1.06E-02/-	1.95E-03/-
+/=/-	0/1/9	0/0/10	0/0/10	3/2/5	1/1/8	0/0/10	0/0/10	2/2/6	2/3/5	0/3/7	0/1/9	2/2/6	2/1/7	0/1/9

**Table 6 biomimetics-09-00399-t006:** The discrete value of the standard modulus.

*x*3: Standard Modulus (mm)						
0.1	0.12	0.15	0.2	0.25	0.3	0.4
0.5	0.6	0.8	0.1	1.25	1.5	2
2.5	3	4	5	6	8	10
12	16	20	25	32	40	50

**Table 7 biomimetics-09-00399-t007:** The results of the SSCGR problem.

Algorithms	*x* _1_	*x* _1_	*x* _1_	*x* _1_	*x* _1_	*x* _1_
CMRLCCOA	84.3521	164.5389	12	76	96	15
KOA	68.31117	163.3201	12	61	87	19
TROA	127.9832	236.9841	6	100	138	23
BWO	125.2962	242.9624	10	77	72	15
AO	52.0544	150.0000	8	68	84	19
HBA	59.0237	150.0234	8	67	83	21
SWO	112.3465	265.799	6	108	148	24
GMO	125.6832	234.0342	5	107	140	27
OMA	120.4782	243.8961	6	100	98	24
GO	120.9341	230.4218	6	108	138	25

**Table 8 biomimetics-09-00399-t008:** The solution quality of SSCGR problem.

Algorithms	Best Cost	Worse Cost	Average Cost	Standard Deviation
CMRLCCOA	9,753,030	19,553,655	15,358,330	258,799.75
KOA	11,703,116	49,804,525	23,770,950	9,668,408.43
TROA	11,020,774	37,884,551	20,846,936	6,707,838.46
BWO	14,902,880.2	17,193,421.69	15,818,208.26	544,502.38
AO	15,235,791.91	16,184,369.46	15,671,961.51	255,292.88
HBA	15,525,037.23	16,184,369.46	15,986,219	282,022.135
SWO	19,478,312.31	28,378,921.72	24,589,213.91	5,423,637.31
GMO	20,786,321.63	34,762,811.82	28,970,643.84	6,272,892.32
OMA	28,876,731.35	34,678,891.78	31,328,901.12	256,893.21
GO	26,895,531.98	36,755,467.90	31,548,903.82	4,983,221.34

**Table 9 biomimetics-09-00399-t009:** Numerical results of ten algorithms for the WBD.

Element	CMRLCCOA	KOA	TROA	BWO	AO	HBA	SWO	GMO	OMA	GO
*x* _1_	0.634	0.163	0.205	0.204	0.156	0.723	0.621	0.169	0.483	0.341
*x* _2_	4.211	4.981	3.572	3.721	5.242	1.502	3.411	4.822	3.274	2.592
*x* _3_	6.802	9.164	9.824	9.311	8.968	5.368	5.391	9.231	8.021	7.608
*x* _4_	0.633	0.246	0.214	0.287	0.219	0.642	0.511	0.205	0.425	0.328
Best	1.660	1.664	1.661	1.663	1.747	2.011	1.683	1.662	1.673	1.691
Worse	1.671	2.130	1.683	2.173	2.162	4.702	6.326	2.750	2.238	2.691
Mean	1.665	1.932	1.674	1.801	1.944	3.164	3.890	2.137	1.902	2.185
Std	0.006	0.132	0.008	0.121	0.133	0.783	1.311	0.185	0.133	0.170

**Table 10 biomimetics-09-00399-t010:** Statistical results of cantilever beam design issues.

Element	CMRLCCOA	KOA	TROA	BWO	AO	HBA	SWO	GMO	OMA	GO
*x* _1_	5.970619	6.340312	5.934021	6.094141	5.952939	6.023157	6.093914	5.873219	6.013591	5.909183
*x* _2_	5.271230	5.329041	5.312115	5.245089	5.279312	5.372044	5.172349	5.309218	5.305521	5.328713
*x* _3_	4.463102	4.502875	4.476823	4.454137	4.47316	4.780241	4.768092	4.457552	4.432802	4.795219
*x* _4_	3.476491	3.592133	3.508861	3.425591	3.469623	3.542192	3.523891	3.492033	3.509931	3.480216
*x_5_*	2.137348	2.160322	2.436287	2.102532	2.149688	2.023797	2.153571	2.437761	2.189233	2.039822
Best	13.302191	16.433288	14.306442	13.32191	13.317692	13.347884	13.863016	13.712833	13.690375	13.926679
Worse	3.313796	24.306681	27.683391	13.396380	13.329414	14.216902	13.926871	16.086629	14.283347	19.37228
Mean	13.308977	20.499271	19.903173	13.358301	13.322805	13.983346	13.890766	14.03799	13.998273	15.349741
Std	3.16E-05	3.8891	4.0399	1.98E-02	2.93E-03	2.57E-02	1.79E-02	9.91E-01	3.88E-02	2.67E-01

**Table 11 biomimetics-09-00399-t011:** Path lengths of CMRLCCOA and other algorithms.

Element	CMRLCCOA	KOA	TROA	BWO	AO	HBA	SWO	GMO	OMA	GO
Length/km	51.231801	57.228763	58.183246	51.243317	51.375423	53.902461	55.320411	51.410963	53.801934	57.944201

## Data Availability

All data generated or analyzed during the study are included in this published article.

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
