# Peer review of "CMRLCCOA: Multi-Strategy Enhanced Coati Optimization Algorithm for Engineering Designs and Hypersonic Vehicle Path Planning"

_biomimetics, 2024, doi:10.3390/biomimetics9070399_

Round 1
Reviewer 1 Report
Comments and Suggestions for Authors
The authors propose an improvement of Coati optimization algorithm with aim to improve algorithm performance and convergence. The authors combine several well known ideas. When some method is used it is more correct some f the works of the person, ho proposed this algorithm, to be cited instead of other author. Thus please replace references from [12] to [19] with publications of the authors of cited methods.
The paper is too long. The list of references can be shorten.
Regarding the tests the results are very similar with different algorithms or SA outperforms others. Thus the proposed algorithm is not the best. How the authors will explain this?
Reviewer 2 Report
Comments and Suggestions for Authors
See the enclosed PDF file for the report.

N/A
Round 2
Reviewer 2 Report
Comments and Suggestions for Authors
The article seems to be suitable for publication now.